# Consistent Diffusion Models: Mitigating Sampling Drift by Learning to be Consistent

**Giannis Daras**[*]
Department of Computer Science
University of Texas at Austin

**Yuval Dagan**[*]
Electrical Engineering and Computer Science
University of California, Berkeley

**Alexandros G. Dimakis**
Department of ECE
University of Texas at Austin

**Constantinos Daskalakis**
Electrical Engineering and Computer Science
Massachusetts Institute of Technology

## Abstract

Imperfect score-matching leads to a shift between the training and the sampling distribution of diffusion models. Due to the recursive nature of the generation process, errors in previous steps yield sampling iterates that drift away from the training distribution. However, the standard training objective via Denoising Score Matching (DSM) is only designed to optimize over non-drifted data. To train on drifted data, we propose to enforce a *Consistency* property (CP) which states that predictions of the model on its own generated data are consistent across time. Theoretically, we show that the differential equation that describes CP together with the one that describes a conservative vector field, have a unique solution given some initial condition. Consequently, if the score is learned well on non-drifted points via DSM (enforcing the true initial condition) then enforcing CP on drifted points propagates true score values. Empirically, we show that enforcing CP improves the generation quality for conditional and unconditional generation on CIFAR-10, and in AFHQ and FFHQ. We open-source our code and models: https://github.com/giannisdaras/cdm.

## 1 Introduction

The diffusion-based [47, 49, 18] approach to generative models has been successful across various modalities, including images [41, 44, 14, 39, 29, 51, 43, 16, 10, 11], videos [19, 20, 21], audio [32], 3D structures [40], proteins [1, 53, 45, 9], and medical applications [23, 3].

Diffusion models generate data by first drawing a sample from a noisy distribution and slowly *denoising* this sample to ultimately obtain a sample from the target distribution. This is achieved by sampling, in reverse from time $t = 1$ down to $t = 0$, a stochastic process $\{x_t\}_{t \in [0,1]}$ wherein $x_0$ is distributed according to the target distribution $p_0$ and, for all $t$,

$$x_t \sim p_t \quad \text{where} \quad p_t := p_0 \oplus \mathcal{N}(0, \sigma_t^2 I_d), \tag{1}$$

where $\oplus$ denotes the convolution operator. That is, $p_t$ is the distribution resulting from corrupting a sample from $p_0$ with noise sampled from $\mathcal{N}(0, \sigma_t^2 I_d)$, where $\sigma_t$ is given by an increasing function of $t$, such that $\sigma_0 = 0$ and $\sigma_1$ is sufficiently large so that $p_1$ is nearly indistinguishable from pure noise. We note that diffusion models have been generalized to other types of corruptions by the recent works

---

[*]These authors contributed equally to this work.

37th Conference on Neural Information Processing Systems (NeurIPS 2023).

of Daras et al. [12], Bansal et al. [4], Hoogeboom and Salimans [22], Deasy et al. [13], Nachmani et al. [38].

In order to sample from a diffusion model, i.e. sample the afore-described process in reverse time, it suffices to know the *score function* $s(x, t) = \nabla_x \log p(x, t)$, where $p(x, t)$ is the density of $x_t \sim p_t$. Indeed, given a sample $x_t \sim p_t$, one can use the score function at $x_t$, i.e. $s(x_t, t)$, to generate a sample from $p_{t-dt}$ by taking an infinitesimal step of a stochastic or an ordinary differential equation [51, 48], or by using Langevin dynamics [17, 50].[2] Hence, in order to train a diffusion model to sample from a target distribution of interest $p_0^*$ it suffices to learn the score function $s^*(x, t)$ using samples from the corrupted distributions $p_t^*$ resulting from $p_0^*$ and a particular noise schedule $\sigma_t$. Notice that those samples can be easily drawn given samples from $p_0^*$.

**The Sampling Drift Challenge:**   Unfortunately, the true score function $s^*(x, t)$ is not perfectly learned during training. Thus, at generation time, the samples $x_t$ drawn using the learned score function, $s_\theta(x, t)$, in the ways discussed above, drift astray in distribution from the true corrupted distributions $p_t^*$. This drift becomes larger for smaller $t$ due to compounding of errors and is accentuated by the fact that the further away a sample $x_t$ is from the likely support of the true $p_t^*$ the larger the error $\|s_\theta(x_t, t) - s^*(x_t, t)\|$ between the learned and the true score function at $x_t$, which feeds into an even larger drift between the distribution of $x_{t'}$ from $p_{t'}^*$ for $t' < t$; see e.g. [46, 18, 39, 6]. These challenges motivate the question:

*Question.* How can one train diffusion models to improve the error $\|s_\theta(x, t) - s^*(x, t)\|$ between the learned and true score function on inputs $(x, t)$ where $x$ is unlikely under the target noisy distribution $p_t^*$?

A direct approach to this challenge is to train our model to minimize the afore-described error on pairs $(x, t)$ where $x$ is sampled from distributions other than $p_t^*$. However, there is no straightforward way to do so, because we do not have direct access to the values of the true score function $s^*(x, t)$.

This motivates us to propose a training method to mitigate sampling drift by enforcing that the learned score function satisfies an invariant, that we call the Consistency Property (CP). This property can be optimized without using any samples from the target distribution $p_0^*$. We will show that theoretically, enforcing CP on drifted points, in conjunction with minimizing the standard score matching objective on non drifted points, suffices to learn the correct score everywhere - at least in the theoretical limit where the error approaches zero and when one also enforces conservative vector field. We also provide experiments illustrating that regularizing the standard score matching objective using our CP improves sample quality. Further, we provide an ablation study that further provides evidence to this phenomenon of score propagation.

**Our Approach:**   The true score function $s^*(x, t)$ is closely related to another function, called the *optimal denoiser*, which predicts a clean sample $x_0 \sim p_0^*$ from a noisy observation $x_t = x_0 + \sigma_t \eta$ where the noise is $\eta \sim \mathcal{N}(0, I_d)$. The optimal denoiser (under the $\ell_2$ loss) is the conditional expectation:

$$h^*(x, t) := \mathbb{E}_{\substack{x_0 \sim p_0^* \\ x_t = x_0 + \sigma_t \eta \\ \eta \sim \mathcal{N}(0, I_d)}} [x_0 \mid x_t = x],$$

and the true score function can be obtained from the optimal denoiser as follows: $s^*(x, t) = (h^*(x, t) - x)/\sigma_t^2$. This result is known as *Tweedie's Formula* [15]. Indeed, the standard training technique, via *score-matching*, explicitly trains for the score through the denoiser $h^*$ [54, 15, 36, 30, 35].

We are now ready to state our Consistency Property (CP). We will say that a (denoising) function $h_\theta(x, t)$ satisfies *CP* iff

$$\mathbb{E}_\theta[x_0 | x_t = x] = h_\theta(x, t), \ \forall t \in [0, 1], \forall x,$$

where the *expectation is with respect to a sample from the **learned** reverse process*, defined in terms of the implied score function $s_\theta(x, t) = (h_\theta(x, t) - x)/\sigma_t^2$, when this is initialized at $x_t = x$ and run backwards in time to sample $x_0$. See Eq. (3) for the precise stochastic differential equation and its justification. In particular, $h_\theta$ satisfies CP if the prediction $h_\theta(x, t)$ of the conditional expectation of the clean image $x_0$ given $x_t = x$ equals the expected value of an image that is generated by the

---

[2]Some of these methods, such as Langevin dynamics, require also to know the score function in the neighborhood of $x_t$.

learned reversed process, starting from $x_t = x$. Equivalently, one can formulate this property as requiring $x_t$ to follow a reverse Martingale (see Lemma 3.1).

While there are several other properties that the score function of a diffusion process must satisfy, e.g. the Fokker-Planck equation [33], our first theoretical result is that the $h_\theta(x, t)$ satisfying the Consistency Property suffices (in conjunction with the conservativeness of its score function $s_\theta(x, t) = (h_\theta(x, t) - x)/\sigma_t^2$) to guarantee that $s_\theta$ must be the score function of a diffusion process (and must thus satisfy any other property that a diffusion process must satisfy). If additionally $s_\theta(x, t)$ equals the score function $s^*(x, t)$ of a target diffusion process at a single time $t = t_0$ and an open subset of $x \in \mathbb{R}^d$, then it equals $s^*$ everywhere. We comment that the formal theorem is proved for an idealistic setting when the error is (or approaches) zero. Still, it is likely to believe that even in the finite-error regime, training with DSM in-sample and enforcing CP off-sample is expected to improve the score function values off-sample. The statement is summarized as follows below:

**Theorem 1.1** (informal). *If some denoiser $h_\theta(x, t)$ satisfies CP and its corresponding score function $s_\theta(x, t) = (h_\theta(x, t) - x)/\sigma_t^2$ is a conservative field, then $s_\theta(x, t)$ is the score function of a diffusion process, i.e. the generation process using score function $s_\theta$, is the inverse of a diffusion process. If additionally $s_\theta(x, t) = s^*(x, t)$ for a single $t = t_0$ and for all $x$ in an open subset of $\mathbb{R}^d$, where $s^*$ is the score function of a target diffusion process, then $s_\theta(x, t) = s^*(x, t)$ everywhere.*

Simply put, the above statement states that: i) satisfying CP and being a conservative vector field is enough to guarantee that the sampling process is the inverse of some diffusion process and ii) to learn the score function everywhere it suffices to learn it for a single $t_0$ and an open subset of $x$'s.

We propose a loss function to train for the Consistency Property and we show experimentally that regularizing the standard score matching objective using our property leads to better models.

**Summary of Contributions:**

1. We identify an invariant property, consistency of the denoiser $h_\theta$, that any perfectly trained model should satisfy.

2. We prove that if the denoiser $h_\theta(x, t)$ satisfies CP and its implied score function $s_\theta(x, t) = (h_\theta(x, t) - x)/\sigma_t^2$ is a conservative field, then $s_\theta(x, t)$ is the score function of *some* diffusion process, even if there are learning errors with respect to the score of the target process, which generates the training data.

3. We prove that optimizing for the score in a subset of the domain and enforcing these two properties, guarantees that the score is learned correctly in all the domain, in the limit where the error approaches zero.

4. We propose a novel training objective that enforces the Consistency Property. Our new objective optimizes the network to have consistent predictions on data points from the *learned* distribution.

5. We show experimentally that, paired with the original Denoising Score Matching (DSM) loss, our objective improves generation quality on conditional and unconditional generation on CIFAR-10, and in AFHQ and FFHQ.

6. We conduct an ablation study which showcases that even if we do not optimize for DSM for some values of $t$, satisfying CP enforces good score approximation there.

## 2   Background

**Diffusion processes, score functions and denoising.**    Diffusion models are trained by solving a supervised regression problem [49, 18]. The function that one aims to learn, called the score function (defined below), is equivalent (up to a linear transformation) to a denoising function [15, 54], whose goal is to denoise an image that was injected with noise. In particular, for some target distribution $p_0$, one's goal is to learn the following function $h \colon \mathbb{R}^d \times [0, 1] \to \mathbb{R}^d$:

$$h(x, t) = \mathbb{E}_{x_0 \sim p_0, \ x_t \sim \mathcal{N}(x_0, \sigma_t^2 I_d)}[x_0 \mid x_t = x]. \tag{2}$$

In other words, the goal is to predict the expected "clean" image $x_0$ given a corrupted version of it, assuming that the image was sampled from $p_0$ and its corruption was done by adding noise to it

from $\mathcal{N}(0, \sigma_t^2 I_d)$, where $\sigma_t^2$ is a non-negative and increasing function of $t$. Given such a function $h$, we can generate samples from $p_0$ by solving a Stochastic Differential Equation (SDE) that depends on $h$ [51]. Specifically, one starts by sampling $x_1$ from some fixed distribution and then runs the following SDE backwards in time:

$$dx_t = -g(t)^2 \frac{h(x_t, t) - x_t}{\sigma_t^2} dt + g(t)d\overline{B}_t, \tag{3}$$

where $\overline{B}_t$ is a reverse-time Brownian motion[3] and $g(t)^2 = \frac{d\sigma_t^2}{dt}$. To explain how Eq. (3) was derived, consider the *forward* SDE that starts with a clean image $x_0$ and slowly injects noise:

$$dx_t = g(t)dB_t, \ x_0 \sim p_0, \tag{4}$$

where $B_t$ is a standard Brownian motion. We notice here that the $x_t$ under Eq. (4) is $\mathcal{N}(x_0, \sigma_t^2 I_d)$, where $x_0 \sim p_0$, so it has the same distribution as in Eq. (2). Remarkably, such SDEs are reversible in time [2]. Hence, the diffusion process of Eq. (4) can be viewed as a reversed-time diffusion:

$$dx_t = -g(t)^2 \nabla_x \log p(x_t, t)dt + g(t)d\overline{B}_t, \tag{5}$$

where $p(x_t, t)$ is the density of $x_t$ at time $t$. We note that $s(x, t) := \nabla_x \log p(x, t)$ is called the *score function* of $x_t$ at time $t$. Using Tweedie's lemma [15], one obtains the following relationship between the denoising function $h$ and the score function:

$$\nabla_x \log p(x, t) = \frac{h(x, t) - x}{\sigma_t^2}. \tag{6}$$

Substituting Eq. (6) in Eq. (5), one obtains Eq. (3).

**Training via denoising score matching.** The standard way to train $h_\theta$ is via *denoising score matching*. This is performed by obtaining samples of $x_0 \sim p_0$ and $x_t \sim \mathcal{N}(x_0, \sigma_t^2 I_d)$ and training to minimize

$$\mathbb{E}_{x_0 \sim p_0, x_t \sim \mathcal{N}(x_0, \sigma_t^2 I_d)} L_{t, x_t, x_0}^{\mathrm{SM}}(\theta) = \mathbb{E}_{x_0 \sim p_0, x_t \sim \mathcal{N}(x_0, \sigma_t^2 I_d)} \|h_\theta(x_t, t) - x_0\|^2,$$

where the optimization is over some family of functions, $\{h_\theta\}_{\theta \in \Theta}$. It was shown by Vincent [54] that optimizing Eq. (2) is equivalent to optimizing $h_\theta$ in mean-squared-error on a random point $x_t$ that is a noisy image, $x_t \sim \mathcal{N}(x_0, \sigma_t^2 I_d)$ where $x_0 \sim p_0$:

$$\mathbb{E}_{x_t} \|h_\theta(x_t, t) - h^*(x_t, t)\|^2,$$

where $h^*$ is the true denoising function from Eq. (2).

## 3 Theory

We define below the Consistency Property that a function $h$ should satisfy. Simply put, it states that the output of $h(x, t)$ (which is meant to approximate the conditional expectation of $x_0$ conditioned on $x_t = x$) is consistent with the average point $x_0$ generated using $h$ and conditioning on $x_t = x$. Recall from the previous section that generation according to $h$ conditioning on $x_t = x$ is done by running the following SDE backwards in time conditioning on $x_t = x$:

$$dx_t = -g(t)^2 \frac{h(x_t, t) - x_t}{\sigma_t^2} dt + g(t)^2 d\overline{B}_t, \tag{7}$$

CP is therefore defined as follows:

*Property* 1 (**Consistency Property**.). A function $h \colon \mathbb{R}^d \times [0, 1] \to \mathbb{R}^d$ is said to satisfy *CP* iff for all $t \in (0, 1]$ and all $x \in \mathbb{R}^d$,

$$h(x, t) = \mathbb{E}_h[x_0 \mid x_t = x], \tag{8}$$

where $\mathbb{E}_h[x_0 \mid x_t = x]$ corresponds to the conditional expectation of $x_0$ in the process that starts with $x_t = x$ and samples $x_0$ by running the SDE of Eq. (7) backwards in time (where note that the SDE uses $h$).

---

[3]A Brownian motion $B_t$ is a stochatic process that injects noise while one goes *forward* in time, and a reverse-time Brownian motion $\bar{B}_t$ injects noise while one goes backwards in time.

The following lemma states that Property 1 holds if and only if the model prediction, $h(x,t)$, $h(x_t, t)$ is a reverse-Martingale under the same process of Eq. (7).

**Lemma 3.1.** *Property 1 holds if and only if the following two properties hold:*

- *The function $h$ is a reverse-Martingale, namely: for all $t > t'$ and for any $x$:*

$$h(x,t) = \mathbb{E}_h[h(x_{t'}, t') \mid x_t = x],$$

*where the expectation is over $x_{t'}$ that is sampled according to Eq. (7) with the same function $h$, given the initial condition $x_t = x$.*

- *For all $x \in \mathbb{R}^d$, $h(x, 0) = x$.*

The proof of this lemma is included in Section B.2. Further, we introduce one more property that will be required for our theoretical results: the learned vector-field should be conservative.

*Property* 2 (**Conservative vector field / Score Property.**). Let $h \colon \mathbb{R}^d \times [0,1] \to \mathbb{R}^d$. We say that $h$ induces a *conservative vector field* (or that satisfies the score property) if for any $t \in (0,1]$ there exists some probability density $p(\cdot, t)$ such that

$$\frac{h(x,t) - x}{\sigma_t^2} = \nabla \log p(x,t).$$

We note that the optimal denoiser, i.e., $h$, defined as in Eq. (2), satisfies both of the properties we introduced. In the paper, we will focus on enforcing CP and we are going to assume conservativeness for our theoretical results. This assumption can be relaxed to hold only at a *single* $t \in (0,1]$ using the results of Lai et al. [33].

Next, we show the theoretical consequences of enforcing Properties 1 and 2. First, we show that this enforces $h$ to indeed correspond to a denoising function, namely, $h$ satisfies Eq. (2) for some distribution $p'_0$ over $x_0$. However, this does not imply that $p_0$ is the *correct* underlying distribution that we are trying to learn. Indeed, these properties can apply to any distribution $p_0$. Yet, we can show that if we learn $h$ correctly for some inputs and if these properties apply everywhere then $h$ is learned correctly everywhere.

**Theorem 3.2.** *Let $h \colon \mathbb{R}^d \times [0,1] \to \mathbb{R}^d$ be a bounded continuous function. Then:*

1. *The function $h$ satisfies both Properties 1 and 2 if and only if $h$ is defined by Eq. (2) for some distribution $p_0$.*

2. *Assume that $h$ satisfies Properties 1 and 2. Further, let $h^*$ be another function that corresponds to Eq. (2) with some initial distribution $p_0^*$. Assume that $h = h^*$ on some open set $U \subseteq \mathbb{R}^d$ and some fixed $t_0 \in (0,1]$, namely, $h(x, t_0) = h^*(x, t_0)$ for all $x \in U$. Then, $h^*(x,t) = h(x,t)$ for all $x$ and all $t$.*

*Remark* 3.3. While our theorem uses Eq. (2) which describes the VE-SDE (Variance Exploding SDE), it is also valid for VP-SDE (Variance Preserving SDE), as these two SDEs are equivalent up to appropriate scaling (see e.g. [31, 28]).

## 4 Method

Theorem 3.2 motivates enforcing CP on the learned model. We notice that the CP Equation Eq. (8) may be expensive to train for, because it requires one to generate whole trajectories. Rather, we use the equivalent Martingale assumption of Lemma 3.1, which can be observed locally with only partial trajectories[4]. We suggest the following loss function, for some fixed $t, t'$ and $x$:

$$L_{t,t',x}^{\mathrm{CP}}(\theta) = \frac{1}{2} \left\| \mathbb{E}_\theta[h_\theta(x_{t'}, t') \mid x_t = x] - h_\theta(x,t) \right\|^2,$$

where the expectation $\mathbb{E}_\theta[\cdot \mid x_t = x]$ is taken according to process Eq. (7) parameterized by $h_\theta$ with the initial condition $x_t = x$. Differentiating this expectation, one gets the following (see Section B.1

---

[4]According to Lemma 3.1, in order to completely train for Property 1, one has to also enforce $h_\theta(x, 0) = x$, however, this is taken care from the denoising score matching objective Eq. (2).

for full derivation):

$$\nabla L_{t,t',x}^{CP}(\theta) = \mathbb{E}_\theta \left[ h_\theta(x_{t'},t') - h_\theta(x_t,t) \mid x_t = x \right]^\top \mathbb{E}_\theta \left[ h_\theta(x_{t'},t')\nabla_\theta \log\left( p_\theta(x_{t'} \mid x_t = x) \right) + \right.$$

$$\left. \nabla_\theta h_\theta(x_{t'},t') - \nabla_\theta h_\theta(x_t,t) \;\middle|\; x_t = x \right],$$

where $p_\theta$ corresponds to the same probability measure where the expectation $\mathbb{E}_\theta$ is taken from and $\nabla_\theta h_\theta$ corresponds to the Jacobian matrix of $h_\theta$ where the derivatives are taken with respect to $\theta$. Notice, however, that computing the expectation accurately might require a large number of samples. Instead, it is possible to obtain a stochastic gradient of this target by taking two samples, $x_{t'}$ and $x'_{t'}$, independently, from the conditional distribution of $x_{t'}$ conditioned on $x_t = x$ and replace each of the two expectations in the formula above with one of these two samples.

We further notice the gradient of the CP loss can be written as

$$\nabla_\theta L_{t,t',x}^{CP}(\theta) = \frac{1}{2} \underbrace{\nabla_\theta \left\| \mathbb{E}_\theta[h_\theta(x_{t'},t')] - h_\theta(x,t) \right\|^2}_{A} +$$

$$\mathbb{E}_\theta \left[ h_\theta(x_{t'},t') - h_\theta(x,t) \right]^\top \mathbb{E}_\theta \left[ \nabla_\theta \log\left( p(x_{t'}) \right) h_\theta(x_{t'},t') \right] \qquad (9)$$

In order to save on computation time, we trained by taking gradient steps with respect to only the first summand (term A) in this decomposition. This term appears in line (4), while we ignored the term in line (9). Notice that if CP is preserved then this term becomes zero, which implies that no update is made, as desired.

It remains to determine how to select $t, t'$ and $x_{t'}$. Notice that $t$ has to vary throughout the whole range of $[0,1]$ whereas $t'$ can either vary over $[0,t]$, however, it is sufficient to take $t' \in [t-\epsilon, t]$. However, the further away $t$ and $t'$ are, we need to run more steps of the reverse SDE to avoid large discretization errors. Instead, we enforce the property only on small time windows using that consistency over small intervals implies global consistency. We notice that $x_t$ can be chosen arbitrarily and two possible choices are to sample it from the target noisy distribution $p_t$ or from the model. See Section D, Algorithm 1 for a pseudocode with more implementation details.

*Remark* 4.1. It is important to sample $x_{t'}$ conditioned on $x_t$ according to the specific SDE Eq. (7). While a variety of alternative SDEs exist which preserve the same marginal distribution at any $t$, they might not preserve the conditionals.

## 5   Experiments

For all our experiments, we rely on the official open-sourced code and the training and evaluation hyper-parameters from the paper "*Elucidating the Design Space of Diffusion-Based Generative Models*" [28] that, to the best of our knowledge, holds the current state-of-the-art on conditional generation on CIFAR-10 and unconditional generation on CIFAR-10, AFHQ ($64 \times 64$ resolution), FFHQ ($64 \times 64$ resolution). We refer to the models trained with our regularization as "CDM (Ours)" and to models trained with vanilla Denoising Score Matching (DSM) as "EDM" models. "CDM" models are trained with the weighted objective:

$$L_\lambda^{ours}(\theta) = \mathbb{E}_t \left[ \mathbb{E}_{x_0 \sim p_0, x_t \sim \mathcal{N}(x_0, \sigma_t^2 I_d)} L_{t,x_t,x_0}^{SM}(\theta) + \lambda \, \mathbb{E}_{x_t \sim p_t} \mathbb{E}_{t' \sim \mathcal{U}[t-\epsilon,t]} L_{t,t',x_t}^{CP}(\theta) \right],$$

while the "EDM" models are trained only with the first term of the outer expectation. We also denote in the name whether the models have been trained with the Variance Preserving (VP) [51, 18] or the Variance Exploding [51, 50, 49], e.g. we write EDM-VP. Finally, for completeness, we also report scores from the models of Song et al. [51], following the practice of the EDM paper. We refer to the latter baselines as "NCSNv3" baselines.

We train diffusion models, with and without our regularization, for conditional generation on CIFAR-10 and unconditional generation on CIFAR-10 and AFHQ ($64 \times 64$ resolution). For the re-trained models on CIFAR-10, we use exactly the same training hyperparameters as in Karras et al. [28] and we verify that our re-trained models match (within $1\%$) the FID numbers mentioned in the paper. For AFHQ, we dropped the batch size from the suggested value of $512$ to $256$ to save on computational

resources, which increased the FID from $1.96$ (reported value) to $2.29$. All models were trained for 200k iterations, as in Karras et al. [28]. Finally, we retrain a baseline model on FFHQ for 150k iterations and we finetune it for 5k steps using our proposed objective.

**Implementation Choices and Computational Requirements.** As mentioned, when enforcing CP, we are free to choose $t'$ anywhere in the interval $[0, t]$. When $t, t'$ are far away, sampling $x'_t$ from the distribution $p^\theta_{t'}(x'_t | x_t)$ requires many sampling steps (to reduce discretization errors). Since this needs to be done for every Gradient Descent update, the training time increases significantly. Instead, we notice that local consistency implies global consistency. Hence, we first fix the number of sampling steps to run in every training iteration and then we sample $t'$ uniformly in the interval $[t - \epsilon, t]$ for some specified $\epsilon$. For all our experiments, we fix the number of sampling steps to 6 which roughly increases the training time needed by 1.5x. We train all our models on a DGX server with 8 A100 GPUs with 80GBs of memory each.

### 5.1 Consistency Property Testing

We are now ready to present our results. The first thing that we check is whether regularizing for CP actually leads to models that are more consistent with their predictions, as the property implies. Specifically, we want to check that the model trained with $L^{\mathrm{ours}}_\lambda$ achieves lower Consistency error, i.e. lower $L^{\mathrm{CP}}_{t,t',x_t}$. To check this, we do the following two tests: i) we fix $t = 1$ and we show how $L^{\mathrm{CP}}_{t,t',x_t}$ changes as $t'$ changes in $[0, 1]$, ii) we fix $t' = 0$ and we show how the loss is changing as you change $t$ in $[0, 1]$. Intuitively, the first test shows how the violation of CP splits across the sampling process and the second test shows how much you finally ($t' = 0$) violate the property if the violation started at time $t$. The results are shown in Figures 1a, 1b, respectively, for the models trained on AFHQ. We include additional results for CIFAR-10, FFHQ in Figures 4, 5, 6, 7 of the Appendix. As shown, indeed regularizing for the CP Loss drops the $L^{\mathrm{CP}}_{t,t',x_t}$ as expected. See Section C.1 for additional details and discussion.

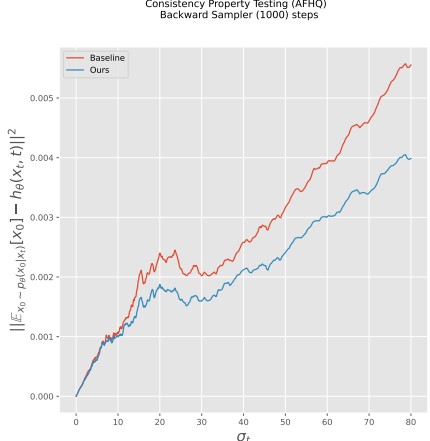

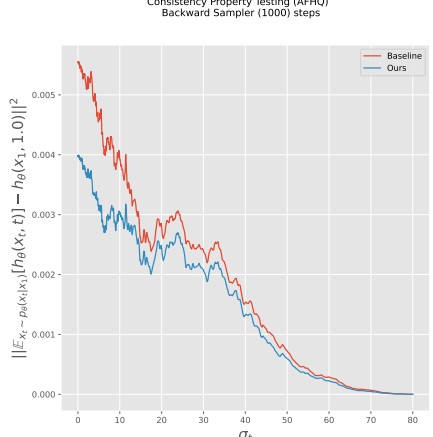

(a) Consistency Property Testing on AFHQ. The plot illustrates how the Consistency Loss, $L^{\mathrm{CP}}_{t,t',x_t}$, behaves for $t' = 0$, as $t$ changes.

(b) Consistency Property Testing on AFHQ. The plot illustrates how the Consistency Loss, $L^{\mathrm{CP}}_{t,t',x_t}$, behaves for $t = 0$, as $t'$ changes.

Figure 1: Consistency Property Testing on AFHQ.

**Performance.** We evaluate the performance of our models. Following the methodology of Karras et al. [28], we generate 150k images from each model and we report the minimum FID computed on three sets of 50k images each. We keep checkpoints during training and we report FID for 30k, 70k, 100k, 150k, 180k and 200k iterations in Table 1. We also report the best FID found for each model, after evaluating checkpoints every 5k iterations (i.e. we evaluate 40 models spanning 200k steps of training). As shown in the Table, the proposed CP regularization yields improvements throughout the training. In the case of CIFAR-10 (conditional and unconditional) where the re-trained baseline was trained with exactly the same hyperparameters as the models in the EDM [28] paper,

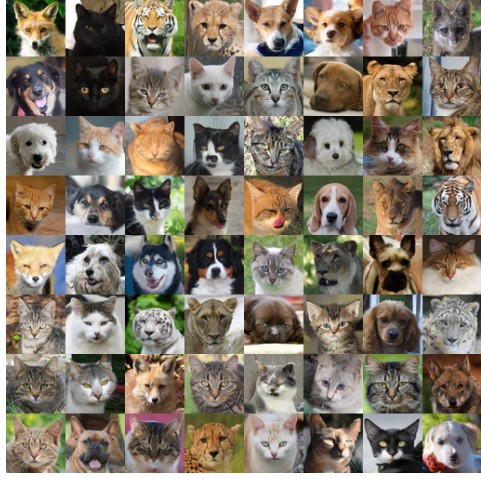

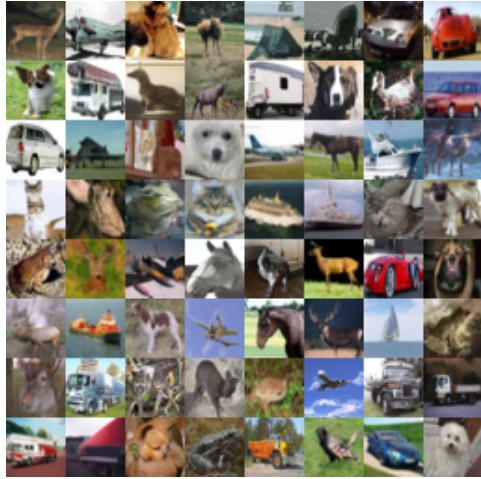

(a) Uncurated images by our model trained on AFHQ. FID: 2.21, NFEs: 79.

(b) Uncurated images by our conditional CIFAR-10 model. FID: 1.77, NFEs: 35.

Figure 2: Comparison of uncurated images generated by two different models.

| Model | | 30k | 70k | 100k | 150k | 180k | 200k | Best |
|---|---|---|---|---|---|---|---|---|
| **CDM-VP (Ours)** | | **3.00** | 2.44 | **2.30** | **2.31** | **2.25** | **2.44** | 2.21 |
| EDM-VP (retrained) | | 3.27 | **2.41** | 2.61 | 2.43 | 2.29 | 2.61 | 2.26 |
| EDM-VP (reported)*[5] | AFHQ | | | | | | | **1.96** |
| EDM-VE (reported)* | | | | | | | | 2.16 |
| NCSNv3-VP (reported)* | | | | | | | | 2.58 |
| NCSNv3-VE (reported)* | | | | | | | | 18.52 |
| **CDM-VP (Ours)** | | **2.44** | **1.94** | **1.88** | 1.88 | **1.80** | **1.82** | **1.77** |
| EDM-VP (retrained) | | 2.50 | 1.99 | 1.94 | **1.85** | 1.86 | 1.90 | 1.82 |
| EDM-VP (reported) | CIFAR-10 | | | | | | | 1.79 |
| EDM-VE (reported) | (cond.) | | | | | | | 1.79 |
| NCSNv3-VP (reported) | | | | | | | | 2.48 |
| NCSNv3-VE (reported) | | | | | | | | 3.11 |
| **CDM-VP (Ours)** | | **2.83** | **2.21** | **2.14** | **2.08** | **1.99** | **2.03** | **1.95** |
| EDM-VP (retrained) | | 2.90 | 2.32 | 2.15 | 2.09 | 2.01 | 2.13 | 2.01 |
| EDM-VP (reported) | CIFAR-10 | | | | | | | 1.97 |
| EDM-VE (reported) | (uncond.) | | | | | | | 1.98 |
| NCSNv3-VP (reported) | | | | | | | | 3.01 |
| NCSNv3-VE (reported) | | | | | | | | 3.77 |
| **CDM-VP (finetuned)** | FFHQ | | | | | | | **2.61** |
| EDM-VP (retrained) | | | | | | | | 2.68 |

Table 1: FID results for deterministic sampling, using the Karras et al. [28] second-order samplers. For the CIFAR-10 models, we do 35 function evaluations and for AFHQ 79.

our CDM models achieve a new state-of-the-art. We further show that our CP regularization can be applied on top of a pre-trained model. Specifically, we train a baseline EDM-VP model on FFHQ $64 \times 64$ for 150k using vanilla Denoising Score Matching. We then do 5k steps of finetuning, with and without our CP regularization and we measure the FID score of both models. The baseline model achieves FID 2.68 while the model finetuned with CP regularization achieves 2.61. This experiment shows the potential of applying our CP regularization to pre-trained models, potentially even at large scale, e.g. we could apply this idea with text-to-image models such as Stable Diffusion [42]. We leave this direction for future work.

Uncurated samples from our best models on AFHQ, CIFAR-10 and FFHQ are given in Figures 2a, 2b and 8. One benefit of the deterministic samplers is the unique identifiability property [51]. Intuitively, this means that by using the same noise and the same deterministic sampler, we can directly compare

visually models that might have been trained in completely different ways. We select a couple of images from Figure 2a (AFHQ generations) and we compare the generated images from our model with the ones from the EDM baseline for the same noises. The results are shown in Figure 3. As shown, the CP regularization fixes several geometric inconsistencies for the picked images. We underline that the shown images are examples for which CP regularization helped and that potentially there are images for which the baseline models give more realistic results.

We note that an appropriate regularization parameter $\lambda$ for our consistency loss has to be chosen. We found that an excessively large $\lambda$ harms the performance: while the CP regularization enforces the model to obey *some* diffusion process, it does not enforce it to obey the *true* diffusion process, and a large $\lambda$ might disturb with the score-matching signal.

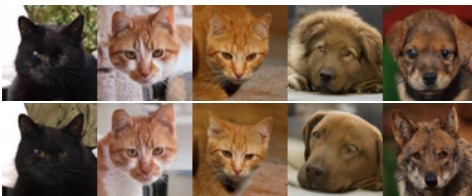

Figure 3: Visual comparison of EDM model (top) and CDM model (Ours, bottom) using deterministic sampling initiated with the same noise. As seen, the CP regularization fixes several geometric inconsistencies and artifacts in the generated images. In order to obtain this comparison, we generated 64 images both with EDM and CDM. We found that in 14 of these images, CDM provided considerable improvement and for the remainder there was no significant difference.

| Model | FID |
|---|---|
| EDM (baseline) | 5.81 |
| CDM, all times $t$ | 5.45 |
| CDM, for some $t$ | 6.59 |
| CDM, for some $t$ early stopped sampling | 14.52 |

Table 2: Ablation study on removing the DSM loss for some $t$. Table reports FID results after 10k steps of training on CIFAR-10.

**Ablation Study for Theoretical Predictions.** One interesting implication of Theorem 3.2 is that it suggests that we only need to learn the score perfectly on some fixed $t_0$ and then the CP implies that the score is learned everywhere (for all $t$ and in the whole space). This motivates the following experiment: instead of using as our loss the weighted sum of DSM and our CP regularization for all $t$, we will not use DSM for $t \leq t_{\text{threshold}}$, for some $t_{\text{threshold}}$ that we test our theory for.

We pick $t_{\text{threshold}}$ such that for $20\%$ of the diffusion (on the side of clean images), we do not train with DSM. For the rest $80\%$ we train with both DSM and our CP regularization. Since this is only an ablation study, we train for only 10k steps on (conditional) CIFAR-10. We report FID numbers for three models: i) training with only DSM, ii) training with DSM and CP regularization everywhere, iii) training with DSM for $80\%$ of times $t$ and CP regularization everywhere. In our reported models, we also include FID of an early stopped sampling of the latter model, i.e. we do not run the sampling for $t < t_{\text{threshold}}$ and we just output $h_\theta(x_{t_{\text{threshold}}}, t_{\text{threshold}})$. The numbers are summarized in Table 2. As shown, the theory is predictive since early stopping the generation at time $t$ gives significantly worse results than continuing the sampling through the times that were never explicitly trained for approximating the score (i.e. we did not use DSM for those times). That said, the best results are obtained by combining DSM and our CP regularization everywhere, which is what we did for all the other experiments in the paper.

## 6   Related Work

The fact that imperfect learning of the score function introduces a shift between the training and the sampling distribution has been well known. Chen et al. [6, 7] analyze how the $l_2$ error in the approximation of the score function propagates to Total Variation distance error bounds between the true and the learned distribution. Several methods for mitigating this issue have been proposed, but the majority of the attempts focus on changing the sampling process [51, 28, 24, 46]. A related work is the Analog-Bits paper [8] that conditions the model during training with past model predictions.

Karras et al. [28] discusses potential violations of invariances, such as the non-conservativity of the induced vector field, due to imperfect score matching. However, they do not formally test or enforce this property. Chao et al. [5] shows that failure to satisfy the conservativity property can harm the performance, and they propose a modification to relieve this degradation. Lai et al. [33] study the problem of regularizing diffusion models to satisfy the Fokker-Planck equation. While we show in Theorem 3.2 that perfect conservative training enforces the Fokker-Planck equation, we notice that their training method is different: they suggest enforcing the equation locally by using the finite differences method to approximate the derivatives. Further, they do not train on drifted data. Instead, we notice that our CP loss is well suited to handle drifted data since it operates across trajectories generated by the model.

A concurrent work by Song et al. [52] proposes Consistency Models (CM), a new class of generative models that output directly the solution of the Probability Flow ODE. This idea resembles the consistency in the model outputs that we enforce through CP, but the motivation is different: CM attempts to accelerate sampling and we attempt to improve generation quality. The two methods are similar in that they enforce a property that the model should satisfy. In the case of the CM, the ODE solver should produce the same solution when evaluated at points that belong to the same trajectory. Our property is that on expectation the predictions should not be changing for points that have the same origin. One way to view it is that CM applies our consistency condition but on the deterministic sampler (for which the expectation becomes the point itself). Subsequent work by Lai et al. [34] compares our CDM to Consistency Models [52] and to Fokker Planck regularization [33].

## 7    Conclusions and Future Work

We proposed an objective that enforces the trained network to follow a reverse Martingale, thereby having self-consistent predictions over time. We optimize this objective with points from the sampling distribution, effectively reducing the sampling drift observed in prior empirical works. Theoretically, we show that CP implies that we are sampling from the reverse of some diffusion process. Together with the assumption that the network has learned the score correctly in a subset of the domain, we can prove that CP (together with conservativity of the vector field) implies that the score is learned correctly everywhere - in the limit where the error approaches zero. Empirically, we use our objective to obtain state-of-the-art for CIFAR-10 and baseline improvements on AFHQ and FFHQ.

There are limitations of our method and several directions for future work. The proposed regularization increases the training time. It would be interesting to explore how to enforce CP in more effective ways in future work. Further, our method does not test nor enforce that the induced vector-field is conservative, which is a key theoretical assumption. Our method guarantees only indirectly improve the performance in the samples from the learned distribution by enforcing some invariant. Finally, our theoretical result holds in the limit where the error of our regularized objective approaches zero and it would be meaningful to theoretically study also the constant-error regime.

## 8    Acknowledgments

This research has been supported by NSF Grants CCF 1763702, AF 1901292, CNS 2148141, Tripods CCF 1934932, IFML CCF 2019844, the Texas Advanced Computing Center (TACC) and research gifts by Western Digital, WNCG IAP, UT Austin Machine Learning Lab (MLL), Cisco and the Archie Straiton Endowed Faculty Fellowship. Giannis Daras has been supported by the Onassis Fellowship, the Bodossaki Fellowship and the Leventis Fellowship. Constantinos Daskalakis has been supported by NSF Awards CCF-1901292, DMS-2022448 and DMS2134108, a Simons Investigator Award, the Simons Collaboration on the Theory of Algorithmic Fairness and a DSTA grant.

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

# A Proof of Theorem 3.2

In Section A.1 we present a proof overview, in Section A.2 we present some preliminaries to the proof, in Section A.3 we include the proof, with proofs of some lemmas omitted and in the remaining sections we prove these lemmas.

## A.1 Proof overview

We start with the first part of the theorem. We assume that $h$ satisfies Properties 1 and 2 and we will show that $h$ is defined by Eq. (2) for some distribution $p_0$ (while the other direction in the equivalence follows trivially from the definitions of these properties). Motivated by Eq. (6), define the function $s \colon \mathbb{R}^d \times (0,1]$ according to

$$s(x,t) = \frac{h(x,t) - x}{\sigma_t^2}. \tag{10}$$

We will first show that $s$ satisfies the partial differential equation

$$\frac{\partial s}{\partial t} = g(t)^2 \left( J_s s + \frac{1}{2} \triangle s \right), \tag{11}$$

where $J_s \in \mathbb{R}^{d \times d}$ is the Jacobian of $s$, $(J_s)_{ij} = \frac{\partial s_i}{x_j}$ and each coordinate $i$ of $\triangle s \in \mathbb{R}^d$ is the Laplacian of coordinate $i$ of $s$, $(\triangle s)_i = \sum_{j=1}^n \frac{\partial^2 s_i}{\partial x_j^2}$. In order to obtain Eq. (11), first, we use a generalization of Ito's lemma, which states that for an SDE

$$dx_t = \mu(x_t, t)dt + g(t)d\overline{B}_t x \tag{12}$$

and for $f \colon \mathbb{R}^d \times [0,1] \to \mathbb{R}^d$, $f(x_t, t)$ satisfies the SDE

$$df(x_t, t) = \left( \frac{\partial f}{\partial t} + J_f \mu - \frac{g(t)^2}{2} \triangle f \right) dt + g(t) J_f d\overline{B}_t.$$

If $f$ is a reverse-Martingale then the term that multiplies $dt$ has to equal zero, namely,

$$\frac{\partial f}{\partial t} + J_f \mu - \frac{g(t)^2}{2} \triangle f = 0.$$

By Lemma 3.1, $h(x_t, t)$ is a reverse Martingale, therefore we can substitute $f = h$ and substitute $\mu = -g(t)^2 s$ according to Eq. (7), to deduce that

$$\frac{\partial h}{\partial t} - g(t)^2 J_h s - \frac{g(t)^2}{2} \triangle h = 0.$$

Substituting $h(x,t) = \sigma_t^2 s(x,t) + x$ according to Eq. (6) yields Eq. (11) as required.

Next, we show that any $s'$ that is the score-function (i.e. gradient of log probability) of some diffusion process that follows the SDE Eq. (4), also satisfies Eq. (11). To obtain this, one can use the Fokker-Planck equation, whose special case states that the density function $p(x,t)$ of any stochastic process that satisfies the SDE Eq. (4) satisfies the PDE

$$\frac{\partial p}{\partial t} = \frac{g(t)^2}{2} \triangle p$$

where $\triangle$ corresponds to the Laplacian operator. Using this one can obtain a PDE for $\nabla_x \log p$ which happens to be exactly Eq. (11) if the process is defined by Eq. (4).

Next, we use Property 2 to deduce that there exists some densities $p(\cdot, t)$ for $t \in [0,1]$ such that

$$s(x,t) = \frac{h(x,t) - x}{\sigma_t^2} = \nabla_x \log p(x,t).$$

Denote by $p'(x,t)$ the score function of the diffusion process that is defined by the SDE of Eq. (4) with the initial condition that $p(x,0) = p'(x,0)$ for all $x$. Denote by $s'(x,t) = \nabla_x \log p'(x,t)$ the score function of $p'$. As we proved above, both $s$ and $s'$ satisfy the PDE Eq. (11) and the same initial

condition at $t = 0$. By the uniqueness of the PDE, it holds that $s(x, t) = s'(x, t)$ for all $t \geq t_0$. Denote by $h^*$ the function that satisfies Eq. (2) with the initial condition $x_0 \sim p_0$. By Eq. (6),

$$s'(x, t) = \frac{h^*(x, t) - x}{\sigma_t^2}.$$

By Eq. (10) and since $s = s'$, it follows that $h = h^*$ and this is what we wanted to prove.

We proceed with proving part 2 of the theorem. We use the notion of an *analytic function* on $\mathbb{R}^d$: that is a function $f \colon \mathbb{R}^d \to \mathbb{R}$ such that at any $x_0 \in \mathbb{R}^d$, the Taylor series of $f$ centered at $x_0$ converges for all $x \in \mathbb{R}^d$ to $f(x)$. We use the property that an analytic function is uniquely determined by its value on any open subset: *If $f$ and $g$ are analytic functions that identify in some open subset $U \subset \mathbb{R}^d$ then $f = g$ everywhere.* We prove this statement in the remainder of this paragraph, as follows: Represent $f$ and $g$ as Taylor series around some $x_0 \in U$. The Taylor series of $f$ and $g$ identify: indeed, these series are functions of the derivatives of $f$ and $g$ which are functions of only the values in $U$. Since $f$ and $g$ equal their Taylor series, they are equal.

Next, we will show that for any diffusion process that is defined by Eq. (4), the probability density of $p(x, t_0)$ at any time $t_0 > 0$ is analytic as a function of $x$. Recall that the distribution of $x_0$ is defined in Eq. (4) as $p_0$ and it holds that the distribution of $x_{t_0}$ is obtained from $p_0$ by adding a Gaussian noise $\mathcal{N}(0, \sigma_t^2 I)$ and its density at any $x$ equals

$$p(x, t_0) = \int_{a \in \mathbb{R}^d} \frac{1}{\sqrt{2\pi}\sigma_{t_0}} \exp\left(-\frac{(x-a)^2}{2\sigma_t^2}\right) dp_0(a).$$

Since the function $\exp(-(x-a)^2/(2\sigma_t^2))$ is analytic, one could deduce that $p(x, t_0)$ is also analytic. Further, $p(x, t_0) > 0$ for all $x$ which implies that there is no singularity for $\log p(x, t_0)$ which can be used to deduce that $\log p(x, t_0)$ is also analytic and further that $\nabla_x \log p(x, t_0)$ is analytic as well.

We use the first part of the theorem to deduce that $s$ is the score function of some diffusion process hence it is analytic. By assumption, $s$ identifies with some target score function $s^*$ in some open subset $U \subseteq \mathbb{R}^d$ at some $t_0$, which, by the fact that $s(x, t_0)$ and $s^*(x, t_0)$ are analytic, implies that $s(x, t_0) = s^*(x, t_0)$ for all $x$. Finally, since $s$ and $s^*$ both satisfy the PDE Eq. (11) and they satsify the same initial condition at $t_0$, it holds that by uniqueness of the PDE $s(x, t) = s^*(x, t)$ for all $x$ and $t$.

## A.2 Preliminaries

**Preliminaries on diffusion processes**   In the next definition we define for a function $F \colon \mathbb{R}^d \to \mathbb{R}^d$ its Jacobian $J_F$, its divergence $\nabla \cdot F$ and its Laplacian $\triangle F$ that is computed separately on each coordinate of $F$:

**Definition A.1.** Given a function $F = (f_1, \ldots, f_n) \colon \mathbb{R}^d \to \mathbb{R}^d$, denote by $J_F \colon \mathbb{R}^d \to \mathbb{R}^{d \times d}$ its Jacobian:

$$(J_F)_{ij} = \frac{\partial f_i(x)}{\partial x_j}.$$

The *divergence* of $F$ is defined as

$$\nabla \cdot F(x) := \sum_{i=1}^n \frac{\partial f_i(x)}{\partial x_i}.$$

Denote by $\triangle F \colon \mathbb{R}^d \to \mathbb{R}^d$ the function whose $i$th entry is the Laplacian of $f_i$:

$$(\triangle F(x))_i = \sum_{j=1}^n \frac{\partial^2 f_i(x)}{\partial x_j^2}.$$

If $F$ is a function of both $x \in \mathbb{R}^d$ and $t \in \mathbb{R}$, then $J_F$, $\triangle f$ and $\nabla \cdot F$ correspond to $F$ as a function of $x$, whereas $t$ is kept fixed. In particular,

$$(J_F(x, t))_{ij} = \frac{\partial f_i(x, t)}{\partial x_j}, \quad (\triangle F(x, t))_i = \sum_{j=1}^n \frac{\partial^2 f_i(x, t)}{\partial x_j^2}, \quad \nabla \cdot F = \sum_{i=1}^n \frac{\partial f_i(x, t)}{\partial x_i}.$$

We use the celebrated Ito's lemma and some of its immediate generalizations:

**Lemma A.2** (Ito's Lemma). *Let $x_t$ be a stochastic process $x_t \in \mathbb{R}^d$, that is defined by the following SDE:*

$$dx_t = \mu(x_t, t)dt + g(t)dB_t,$$

*where $B_t$ is a standard Brownian motion. Let $f \colon \mathbb{R}^d \times \mathbb{R} \to \mathbb{R}$. Then,*

$$df(x_t, t) = \left( \frac{df}{dt} + \nabla_x f^\top \mu(x_t, t) + \frac{g(t)^2}{2} \triangle f \right) dt + g(t) \nabla_x f^\top dB_t.$$

*Further, if $F \colon \mathbb{R}^d \times \mathbb{R} \to \mathbb{R}^d$ is a multi-valued function, then*

$$dF(x_t, t) = \left( \frac{dF}{dt} + J_F \mu + \frac{g(t)^2}{2} \triangle F \right) dt + g(t) J_F dB_t.$$

*Lastly, if $x_t$ is instead defined with a reverse noise,*

$$dx_t = \mu(x_t, t)dt + g(t)d\overline{B}_t,$$

*then the multi-valued Ito's lemma is modified as follows:*

$$dF(x_t, t) = \left( \frac{dF}{dt} + J_F \mu - \frac{g(t)^2}{2} \triangle F \right) dt + g(t) J_F d\overline{B}_t. \tag{13}$$

Lastly, we present the Fokker-Planck equation which states that the probability distribution that corresponds to diffusion processes satisfy a certain partial differential equation:

**Lemma A.3** (Fokker-Planck equation). *Let $x_t$ be defined by*

$$dx_t = \mu(x_t, t)dt + g(t)dB_t,$$

*where $x_t, \mu(x, t) \in \mathbb{R}^d$ and $B_t$ is a Brownian motion in $\mathbb{R}^d$. Denote by $p(x, t)$ the density at point $x$ on time $t$. Then,*

$$\frac{\partial}{\partial t} p(x, t) = -\nabla \cdot (\mu(x, t)p(x, t)) + \frac{g(t)^2}{2} \triangle p(x, t) = -p\nabla \cdot \mu - \mu \nabla \cdot p + \frac{g(t)^2}{2} \triangle p.$$

### Preliminaries on analytic functions

**Definition A.4.** A function $f \colon \mathbb{R}^d \to \mathbb{R}$ is analytic on $\mathbb{R}^d$ if for any $x_0, x \in \mathbb{R}^d$, the Taylor series of $f$ around $x_0$, evaluated at $x$, converges to $f(x)$. We say that $F = (f_1, \ldots, f_n) \colon \mathbb{R}^d \to \mathbb{R}^d$ is an analytic function if $f_i$ is analytic for all $i \in \{1, \ldots, n\}$.

The following holds:

**Lemma A.5.** *If $F, G \colon \mathbb{R}^d \to \mathbb{R}^d$ are two analytic functions and if $F = G$ for all $x \in U$ where $U \subseteq \mathbb{R}^d$, $U \neq 0$, is an open set, then $F = G$ on all $\mathbb{R}^d$.*

This is a well known result and a proof sketch was given in Section 3.

**The heat equation.** The following is a Folklore lemma on the uniqueness of the solutions to the heat equation:

**Lemma A.6.** *Let $p$ and $p'$ be two continuous functions on $\mathbb{R}^d \times [t_0, 1]$ that satisfy the heat equation*

$$\frac{\partial p}{\partial t} = \frac{g(t)^2}{2} \triangle p. \tag{14}$$

*Further, assume that $p(\cdot, t_0) = p'(\cdot, t_0)$. Then, $p = p'$ for all $t \in [t_0, 1]$.*

## A.3 Main proof

In what appears below we denote

$$s(x, t) := \frac{h(x, t) - x}{\sigma_t^2}. \tag{15}$$

We start by claiming that if $h$ satisfies Property 1, then $s$ satisfies the PDE Eq. (11): (proof in Section A.4)

**Lemma A.7.** *Let $h$ satisfy Property 1 and define $s$ according to Eq. (15). Then, $s$ satisfies Eq. (11).*

Next, we claim that the score function of any diffusion process satisfies the PDE Eq. (11): (proof in Section A.5)

**Lemma A.8.** *Let $s$ be the score function of some diffusion process that is defined by Eq. (4). Then, $s$ satisfies the PDE Eq. (11).*

To complete the first part of the proof, denote by $p(\cdot, t)$ the probability distribution such that $s(x, t) = \nabla \log p(x, t)$, whose existence follows from Property 2. We would like to argue that $\{p(\cdot, t)\}_{t \in (0,1]}$ corresponds the probability density of the diffusion

$$dx_t = g(t)dB_t. \tag{16}$$

It suffices to show that for any $t_0 > 0$, $\{p(\cdot, t)\}_{t \in (t_0, 1]}$ corresponds to the same diffusion. To show the latter, let $t_0 \in (0, 1)$ and consider the diffusion process according to Eq. (16) with the initial condition that $x_{t_0} \sim p(\cdot, t_0)$. Denote its score function by $s'$ and notice that it satisfies the PDE Eq. (11) and the initial condition $s'(x, t_0) = \nabla_x \log p(x, t_0) = s(x, t_0)$, where the first equality follows from the definition of a score function and the second from the construction of $p(x, t_0)$. Further, recall that $s(x, t)$ satisfies the same PDE Eq. (11) by Lemma A.4. Next we will show that $s = s'$ for all $t \in [t_0, 1]$, and this will follow from the following lemma: (proof in Section A.6)

**Lemma A.9.** *Let $s$ and $s'$ be two solutions for the PDE (11) on the domain $\mathbb{R}^d \times [t_0, 1]$ that satisfy the same initial condition at $t_0$: $s(x, t_0) = s'(x, t_0)$ for all $x$. Further, assume that for all $t \in [t_0, 1]$ there exist probability densities $p(\cdot, t)$ and $p'(\cdot, t)$ such that $s(x, t) = \nabla_x \log p(x, t)$ and $s'(x, t) = \nabla_x \log p'(x, t)$ for all $x$. Then, $s = s'$ on all of $\mathbb{R}^d \times [t_0, 1]$.*

Then, by uniqueness of the PDE one obtains that $s = s'$ for all $t \in [t_0, 1]$. Hence, $s$ is the score of a diffusion for all $t \geq t_0$ and this holds for any $t_0 > 0$, hence this holds for any $t > 0$. This concludes the proof of the first part of the theorem.

For the second part, let $s^*$ denote some score function of a diffusion process that satisfies Eq. (4). Assume that for some $t_0 > 0$ and some open subset $U \subseteq \mathbb{R}^d$, $s = s^*$, namely $s(x, t_0) = s^*(x, t_0)$ for all $t_0 > 0$ and all $x \in U$. First, we would like to argue that if $s(x, t)$ is the score function of some diffusion process that satisfies Eq. (4), then for any $t_0 > 0$ it holds that $s(x, t_0)$ is an analytic function (proof in Section A.7)

**Lemma A.10.** *Let $x_t$ obey the SDE Eq. (4) with the initial condition $x_0 \sim \mu_0$. Let $t > 0$ and let $s(x, t)$ denote the score function of $x_t$, namely, $s(x, t) = \nabla_x \log p(x, t)$ where $p(x, t)$ is the density of $x_t$. Assume that $\mu_0$ is a bounded-support distribution. Then, $s(x, t)$ is an analytic function.*

Since both $s$ and $s^*$ are scores of diffusion processes, then $s(x, t_0)$ and $s^*(x, t_0)$ are analytic functions. Using the fact that $s = s^*$ on $U \times \{t_0\}$ and using Lemma A.5 we derive that $s(x, t_0) = s^*(x, t_0)$ for all $x$. Let $p$ and $p^*$ denote the densities that correspond to the score functions $s$ and $s^*$ and by definition of a score function, we obtain that for all $x$,

$$\nabla \log p(x, t_0) = s(x, t_0) = s^*(x, t_0) = \nabla \log p^*(x, t_0),$$

which implies, by integration, that

$$\log p(x, t_0) = \log p^*(x, t_0) + c$$

for some constant $c \in \mathbb{R}$. However, $c = 0$. Indeed,

$$1 = \int p(x, t_0)dx = \int e^{\log p(x, t_0)}dx = \int e^{\log p^*(x, t_0) + c}dx = \int p^*(x, t_0)e^c dx = e^c,$$

which implies that $c = 0$ as required. As a consequence, the following lemma implies that $p(x, 0) = p^*(x, 0)$ for all $x$ (proof in Section A.8):

**Lemma A.11.** *Let $x_t$ and $y_t$ be stochastic processes that follow Eq. (4) with initial conditions $x_0 \sim \mu_0$ and $y_0 \sim \mu'_0$ and assume that $\mu_0$ and $\mu'_0$ are bounded-support. Assume that for some $t_0 > 0$, $x_{t_0}$ and $y_{t_0}$ have the same distribution. Then, $\mu_0 = \mu'_0$.*

Without loss of generality, one can replace 0 with any $\tilde{t} \in (0, t_0)$, to obtain that $p(x, \tilde{t}) = p^*(x, \tilde{t})$ for any $\tilde{t} \in [0, t_0]$. Now, $p(x, t_0)$ is analytic, from Lemma A.5, hence it is continuous. Consequently, Lemma A.6 implies that $p = p^*$ in $\mathbb{R}^d \times [t_0, 1]$. This concludes that $p = p^*$ in all the domain, which implies that $s = \nabla \log p = \nabla \log p^* = s^*$, as required.

## A.4 Proof of Lemma A.7

We use Ito's lemma, and in particular Eq. (13), to get a PDE for the function $h(x_t, t)$ where $x_t$ satisfies the stochastic process

$$dx_t = -g(t)^2 s(x_t, t)dt + g(t)d\bar{B}_t.$$

Ito's formula yields that

$$dh(x_t, t) = \left(\frac{\partial h}{\partial t} - g(t)^2 J_F s - \frac{g(t)^2}{2}\triangle h\right)dt + \sigma J_h d\bar{B}_t.$$

Since $(h, s)$ satisfies Property 1 and using Lemma 3.1, $h$ is a reverse martingale which implies that the term that multiplies $dt$ has to equal zero. In particular, we have that

$$\frac{\partial h}{\partial t} - g(t)^2 J_h s - \frac{g(t)^2}{2}\triangle h = 0. \tag{17}$$

By Eq. (15),

$$s = \frac{h - x}{\sigma_t^2}.$$

Therefore,

$$h = x + \sigma_t^2 s.$$

Substituting this in Eq. (17) and using the relation $d\sigma_t^2/dt = g(t)^2$ that follows from Eq. (15), one obtains that

$$\begin{aligned}
0 &= \frac{\partial}{\partial t}(x + \sigma_t^2 s) - g(t)^2 J_{x+\sigma_t^2 s}s - \frac{g(t)^2}{2}\triangle(x + \sigma_t^2 s) \\
&= g(t)^2 s + \sigma_t^2\frac{\partial s}{\partial t} - g(t)^2(I + \sigma_t^2 J_s)s - \frac{g(t)^2\sigma_t^2}{2}\triangle s \\
&= \sigma_t^2\frac{\partial s}{\partial t} - g(t)^2\sigma_t^2 J_s s - \frac{g(t)^2\sigma_t^2}{2}\triangle s.
\end{aligned}$$

Dividing by $\sigma_t^2$, we get that

$$\frac{\partial s}{\partial t} - g(t)^2 J_s s - \frac{g(t)^2}{2}\triangle s = 0,$$

which is what we wanted to prove.

## A.5 Proof of Lemma A.8

We present as a consequence of the Fokker-Plank equation (Lemma A.3) a PDE for the log density $\log p$:

**Lemma A.12.** *Let $x_t$ be defined by*

$$dx_t = \mu(x_t, t)dt + g(t)dB_t.$$

*Then,*

$$\frac{\partial \log p}{\partial t} = -\nabla \cdot \mu - \mu\nabla \cdot \log p + \frac{g(t)^2\|\nabla \log p\|^2}{2} + \frac{g(t)^2\triangle \log p}{2}$$

*Proof.* We would like to replace the partial derivatives of $p$ that appear in Lemma A.3 with the partial derivatives of $\log p$. Using the formula

$$\frac{\partial \log p}{\partial t} = \frac{1}{p}\frac{\partial p}{\partial t},$$

one obtains that

$$\frac{\partial p}{\partial t} = p\frac{\partial \log p}{\partial t}.$$

Similarly,

$$\frac{\partial p}{\partial x_i} = p\frac{\partial \log p}{\partial x_i} \tag{18}$$

which also implies that

$$\nabla p = p\nabla \log p, \quad \nabla \cdot p = p\nabla \cdot \log p.$$

Differentiating Eq. (18) again with respect to $x_i$ and applying Eq. (18) once more, one obtains that

$$\frac{\partial^2 p}{\partial x_i^2} = \frac{\partial}{\partial x_i}\left(p\frac{\partial \log p}{\partial x_i}\right) = \frac{\partial p}{\partial x_i}\frac{\partial \log p}{\partial x_i} + p\frac{\partial^2 \log p}{\partial x_i^2} = p\left(\left(\frac{\partial \log p}{\partial x_i}\right)^2 + \frac{\partial^2 \log p}{\partial x_i^2}\right).$$

Summing over $i$, one obtains that

$$\triangle p = p\sum_{i=1}^{n}\left(\left(\frac{\partial \log p}{\partial x_i}\right)^2 + \frac{\partial^2 \log p}{\partial x_i^2}\right) = p\|\nabla \log p\|^2 + p\triangle \log p. \tag{19}$$

Substituting the partials derivatives of $p$ inside the Fokker-Planck equation in Lemma A.3, one obtains that

$$p\frac{\partial \log p}{\partial t} = -p\nabla \cdot \mu - \mu(p\nabla \cdot \log p) + \frac{g(t)^2}{2}\left(p\|\nabla \log p\|^2 + p\triangle \log p\right).$$

Dividing by $p$, one gets that

$$\frac{\partial \log p}{\partial t} = -\nabla \cdot \mu - \mu\nabla \cdot \log p + \frac{g(t)^2\|\nabla \log p\|^2}{2} + \frac{g(t)^2\triangle \log p}{2}.$$

as required. $\qquad\square$

We are ready to prove Lemma A.8: Substituting $\mu = 0$ in Lemma A.12, one obtains that

$$\frac{\partial \log p}{\partial t} = \frac{g(t)^2\|\nabla \log p\|^2}{2} + \frac{g(t)^2\triangle \log p}{2}.$$

Taking the gradient with respect to $x$, one obtains that

$$\nabla\frac{\partial \log p}{\partial t} = \frac{g(t)^2\nabla\|\nabla \log p\|^2}{2} + \frac{g(t)^2\nabla\triangle \log p}{2}. \tag{20}$$

Since $\partial/\partial x_i$ commutes with $\partial/\partial t$, it holds that

$$\nabla\frac{\partial \log p}{\partial t} = \frac{\partial}{\partial t}\nabla \log p = \frac{\partial s}{\partial t}, \tag{21}$$

recalling that by definition $s = \nabla \log p$. Further,

$$\frac{\partial}{\partial x_i}\|\nabla \log p\|^2 = \sum_{j=1}^{n}\frac{\partial}{\partial x_i}\left(\frac{\partial \log p}{\partial x_j}\right)^2 = 2\sum_{j=1}^{n}\frac{\partial^2 \log p}{\partial x_i\partial x_j}\frac{\partial \log p}{\partial x_j} = 2(H_{\log p}\nabla \log p)_i,$$

where for any function $f\colon \mathbb{R}^d \to \mathbb{R}$, $H_f$ is the Hessian function of $f$ that is defined by

$$(H_f)_{ij} = \frac{\partial^2 f}{\partial x_i\partial x_j}$$

This implies that

$$\nabla\|\nabla \log p\|^2 = 2H_{\log p}\nabla \log p.$$

Further, notice that

$$H_f = J_{\nabla f},$$

which implies that

$$\nabla\|\nabla \log p\|^2 = 2J_{\nabla \log p}\nabla \log p = 2J_s s. \tag{22}$$

Lastly, we get that by the commutative property of partial derivatives,

$$\nabla\triangle \log p = \triangle\nabla \log p = \triangle s. \tag{23}$$

Substituting Eq. (21), Eq. (22) and Eq. (23) in Eq. (20), one obtains that

$$\frac{\partial s}{\partial t} = g(t)^2 J_s s + \frac{g(t)^2\triangle s}{2},$$

as required.

## A.6 Proof of Lemma A.9

We will prove that $p$ and $p'$ satisfy the same PDE (which is the heat equation). Recall that $s$ and $s'$ satisfy

$$\frac{\partial s}{\partial t} = g(t)^2 \left( J_s s + \frac{1}{2} \triangle s \right) = \frac{g(t)^2}{2} \left( \nabla \|s\|^2 + \triangle s \right)$$

By substituting $s = \nabla \log p$,

$$\frac{\partial \nabla \log p}{\partial t} = \frac{g(t)^2}{2} \left( \nabla \|\nabla \log p\|^2 + \triangle \nabla \log p \right).$$

By exchanging the order of derivatives, we obtain that

$$\nabla \frac{\partial \log p}{\partial t} = \nabla \frac{g(t)^2}{2} \left( \|\nabla \log p\|^2 + \triangle \log p \right).$$

By integrating, this implies that

$$\frac{\partial \log p}{\partial t} = \frac{g(t)^2}{2} \left( \|\nabla \log p\|^2 + \triangle \log p \right) + c(t),$$

where $c(t)$ depends only on $t$. Eq. (19) shows that

$$\triangle \log p = \frac{\triangle p}{p} - \|\nabla \log p\|^2.$$

By substituting this in the equation above, we obtain that

$$\frac{\partial \log p}{\partial t} = \frac{g(t)^2}{2} \frac{\triangle p}{p} + c(t).$$

By multiplying both sides with $p$, we get that

$$\frac{\partial p}{\partial t} = p \frac{\partial \log p}{\partial t} = \frac{g(t)^2}{2} \triangle p + c(t). \tag{24}$$

Since $p$ is a probability distribution,

$$\int_{\mathbb{R}^d} p(x, t) dx = 1,$$

therefore,

$$\int \frac{\partial p(x, t)}{\partial t} dx = \frac{\partial}{\partial t} \int_{\mathbb{R}^d} p(x, t) dx = \frac{\partial 1}{\partial t} = 0.$$

Integrating over Eq. (24) we obtain that

$$0 = \int \frac{g(t)^2}{2} \triangle p + c(t) dx = 0 + \int c(t) dx,$$

where the last equation holds since the integral of a Laplacian of probability density integrates to $0$. It follows that $c(t) = 0$ which implies that

$$\frac{\partial p}{\partial t} = \frac{g(t)^2}{2} \triangle p, \tag{25}$$

and the same PDE holds where $p'$ replaces $p$, and this follows without loss of generality. Further, since $\log p$ and $\log p'$ are differentiable, it holds that $p(\cdot, t)$ and $p'(\cdot, t)$ are continuous for all fixed $t$. This implies that $p$ and $p'$ are continuous as functions of $x$ and $t$ since they both satisfy the heat equation Eq. (14). Consequently, Lemma A.6 implies that $p = p'$ on $\mathbb{R}^d \times [t_0, 1]$. Finally, $s = \nabla \log p = \nabla \log p' = s'$, as required.

## A.7 Proof of Lemma A.10

First, recall that since $x_t$ satisfies Eq. (4) with the initial condition $x_0 \sim \mu_0$, then $x_t \sim \mu_0 + \mathcal{N}(0, \sigma_t^2 I)$, namely, $x_t$ is the addition of a random variable drawn from $\mu_0$ and an independent Gaussian $\mathcal{N}(0, \sigma_t^2 I)$. Therefore, the density of $x_t$, which we denote by $p(x, t)$, equals

$$p(x, a) = \mathbb{E}_{a \sim \mu_0} \left[ \frac{1}{\sqrt{2\pi}\sigma_t} \exp \left( -\frac{\|x - a\|^2}{2\sigma_t^2} \right) \right].$$

Using the equation

$$\nabla_x \log f(x) = \frac{\nabla_x f(x)}{f(x)},$$

we get that

$$s(x, a) = \nabla_x \log p(x, a) = \frac{\nabla_x p(x, a)}{p(x, a)} = \frac{\mathbb{E}_{a \sim \mu_0} \left[ \frac{1}{\sqrt{2\pi}\sigma_t} \frac{x - a}{\sigma_t^2} \exp \left( -\frac{\|x - a\|^2}{2\sigma_t^2} \right) \right]}{\mathbb{E}_{a \sim \mu_0} \left[ \frac{1}{\sqrt{2\pi}\sigma_t} \exp \left( -\frac{\|x - a\|^2}{2\sigma_t^2} \right) \right]} \tag{26}$$

By using the fact that the Taylor formula for $e^x$ equals

$$e^x = \sum_{i=0}^{\infty} \frac{e^i}{i!},$$

we obtain that the right hand side of Eq. (26) equals

$$\frac{\mathbb{E}_{a \sim \mu_0} \left[ \frac{1}{\sqrt{2\pi}\sigma_t} \frac{x - a}{\sigma_t^2} \sum_{i=0}^{\infty} \frac{(-1)^i}{i!} \left( \frac{\|x - a\|^2}{2\sigma_t^2} \right)^i \right]}{\mathbb{E}_{a \sim \mu_0} \left[ \frac{1}{\sqrt{2\pi}\sigma_t} \sum_{i=0}^{\infty} \frac{(-1)^i}{i!} \left( \frac{\|x - a\|^2}{2\sigma_t^2} \right)^i \right]} = \frac{\mathbb{E}_{a \sim \mu_0} \left[ \frac{x - a}{\sigma_t^2} \sum_{i=0}^{\infty} \frac{(-1)^i}{i!} \left( \frac{\|x - a\|^2}{2\sigma_t^2} \right)^i \right]}{\mathbb{E}_{a \sim \mu_0} \left[ \sum_{i=0}^{\infty} \frac{(-1)^i}{i!} \left( \frac{\|x - a\|^2}{2\sigma_t^2} \right)^i \right]} \tag{27}$$

We will use the following property of analytic functions: if $f$ and $g$ are analytic functions over $\mathbb{R}^d$ and $g(x) \neq 0$ for all $x$ then $f/g$ is analytic over $\mathbb{R}^d$. Since the denominator at the right hand side of Eq. (27) is nonzero, it suffices to prove that the numerator and the denominator are analytic. We will prove for the denominator and the proof for the numerator is nearly identical. By assumption of this lemma, the support of $\mu_0$ is bounded, hence there is some $M > 0$ such that $\|x\| \leq M$ for any $x$ in the support. Then,

$$\left| \frac{(-1)^i}{i!} \left( \frac{\|x - a\|^2}{2\sigma_t^2} \right)^i \right| \leq \frac{1}{i!} \left( \frac{x^2 + a^2}{\sigma_t^2} \right)^i = \frac{M^{2i}}{\sigma_t^{2i} i!}.$$

This bound is independent on $a$, and summing these absolute values of coefficients for $i \in \mathbb{N}$, one obtains a convergent series. Hence we can replace the summation and the expectation in the denominator at the right hand side of Eq. (27) to get that it equals

$$\sum_{i=0}^{\infty} \frac{(-1)^i}{i!} \mathbb{E}_{a \sim \mu_0} \left[ \left( \frac{\|x - a\|^2}{2\sigma_t^2} \right)^i \right]. \tag{28}$$

This is the Taylor series around 0 of the above-described denominator it converges to the value of the denominator at any $x$. While this Taylor series is taken around 0, we note the Taylor series around any other point $x_0 \in \mathbb{R}^n$ converges as well. This can be shown by shifting the coordinate system by a constant vector such that $x_0$ shifts to 0 and applying the same proof. One deduces that the Taylor series for the denominator around any point $x_0$ converges on all $\mathbb{R}^d$, which implies that the denominator in the right hand side of Eq. (27) is analytic. The numerator is analytic as well by the same argument. Therefore the ratio, which equals $s(x, t)$, is analytic as well as required.

## A.8 Proof of Lemma A.11

Let $t > 0$, denote by $\mu_t$ and $\mu_t'$ the distributions of $x_t$ and $x_t'$, respectively, and by $p(x, t)$ and $p'(x, t)$ the densities of these variables. Then, $\mu_t = \mu_0 + \mathcal{N}(0, \sigma_t^2 I)$, namely, $\mu_t$ is obtained by adding an independent sample from $\mu_0$ with an independent $\mathcal{N}(0, \sigma_t^2 I)$ variable, and similarly for $\mu_t'$. Hence, the density $p(x, t)$ is the convolution of the densities $p(x, 0)$ with the density of a Gaussian $\mathcal{N}(0, \sigma_t^2 I)$.

Denote by $\hat{p}(y,t)$ the Fourier transform of the density $p(x,t)$ with respect to $x$ (while keeping $t$ fixed) and similarly define $\hat{p}'$ as the Fourier transform of $p'$. Denote by $g$ and by $\hat{g}$ the density of $\mathcal{N}(0, \sigma_t^2 I)$ and its Fourier transform, respectively. Denote the convolution of two functions by the operator $*$. Then,

$$p(x,t) = p(x,0) * g(x), \quad p'(x,t) = p'(x,0) * g(x).$$

Since the Fourier transform turns convolutions into multiplications, one obtains that

$$\hat{p}(y,t) = \hat{p}(y,0)\hat{g}(y), \quad \hat{p}'(y,t) = \hat{p}'(y,0)\hat{g}(y).$$

Since $p(x,t) = p'(x,t)$ we obtain that $\hat{p}(y,t) = \hat{p}'(y,t)$. Consequently,

$$\hat{p}(y,0)\hat{g}(y) = \hat{p}'(y,0)\hat{g}(y)$$

Since the Fourier transform of a Gaussian is nonzero, we can divide by $\hat{g}(y)$ to get that

$$\hat{p}(y,0) = \hat{p}'(y,0).$$

This implies that the Fourier transform of $p(x,0)$ equals that of $p'(x,0)$ hence $p(x,0) = p'(x,0)$ for all $x$, as required.

# B  Other proofs

## B.1  Differentiating the loss function

Denote our parameter space as $\Theta \subseteq \mathbb{R}^m$. In order to differentiate $L_{t,t',x}^{\mathrm{SM}}(\theta)$ with respect to $\theta \in \Theta$, we make the following calculations below, and we notice that $\mathbb{E}_\theta$ is used to denote an expectation with respect to the distribution of $x_{[t',t]}$ according to Eq. (7) with $s = s_\theta$ and the initial condition $x_t = x$. In other words, the expectation is over $x_{[t',t]}$ that is taken with respect to the sampler that is parameterized by $\theta$ with the initial condition $x_t = x$. We denote by $p_\theta(x_{[t',t]} \mid x_t = x)$ the corresponding density of $x_{[t',t]}$. For any function $f = (f_1, \ldots, f_n) \colon \Theta \to \mathbb{R}^n$, denote by $\nabla_\theta f$ the Jacobian matrix of $f$, where

$$(\nabla_\theta f)_{i,j} = \frac{\partial f_i}{\partial \theta_j}.$$

For notational consistency, if $f$ is a single-valued function, namely, if $n = 1$, then $\nabla_\theta f$ is a column vector. We begin with the following:

$$
\begin{aligned}
\nabla_\theta \mathbb{E}_\theta \left[ h_\theta(x_{t'}, t') \right] &= \nabla_\theta \int_{\mathbb{R}^d} h_\theta(x_{t'}, t') p_\theta(x_{[t',t]} \mid x_t = x) dx_{t'} \\
&= \int_{\mathbb{R}^d} \nabla_\theta h_\theta(x_{t'}, t') p_\theta(x_{[t',t]} \mid x_t = x) dx_{t'} + \int_{\mathbb{R}^d} h_\theta(x_{t'}, t') \nabla_\theta p_\theta(x_{[t',t]} | x_t = x) dx_{t'} \\
&= \mathbb{E}_\theta \left[ \nabla_\theta h_\theta(x_{t'}, t') \right] + \mathbb{E}_\theta \left[ h_\theta(x_{t'}, t') \frac{\nabla_\theta p_\theta(x_{[t',t]} | x_t = x)}{p_\theta(x_{[t',t]} | x_t = x)} \right] \\
&= \mathbb{E}_\theta \left[ \nabla_\theta h_\theta(x_{t'}, t') \right] + \mathbb{E}_\theta \left[ h_\theta(x_{t'}, t') \nabla_\theta \log \left( p_\theta(x_{[t',t]} \mid x_t = x) \right) \right]
\end{aligned}
$$

Differentiating the whole loss, we get the following:

$$
\begin{aligned}
\nabla_\theta L_{t,t',x}^{\mathrm{SM}}(\theta) &= \frac{1}{2} \nabla_\theta \left( \mathbb{E}_\theta[h_\theta(x_{t'}, t')] - h_\theta(x,t) \right)^2 \\
&= \left( \mathbb{E}_\theta[h_\theta(x_{t'}, t')] - h_\theta(x,t) \right)^\top \left( \nabla_\theta \mathbb{E}[h_\theta(x_{t'}, t')] - \nabla_\theta h_\theta(x,t) \right) \\
&= \mathbb{E}_\theta \left[ h_\theta(x_{t'}, t') - h_\theta(x,t) \right]^\top \mathbb{E}_\theta \left[ \nabla_\theta h_\theta(x_{t'}, t') - \nabla_\theta h_\theta(x,t) \right] \\
&\quad + \mathbb{E}_\theta \left[ h_\theta(x_{t'}, t') - h_\theta(x,t) \right]^\top \mathbb{E}_\theta \left[ h_\theta(x_{t'}, t') \nabla_\theta \log \left( p_\theta(x_{[t',t]} \mid x_t = x) \right) \right]
\end{aligned}
$$

Let us compute the gradient of the log density. We use the discrete process, and let us assume that $t = t_0 > t_1 > \cdots > t_k = t'$ are the sampling times. Then,

$$p_\theta(x_{[t',t]} \mid x_t = x) = \prod_{i=1}^{k} p_\theta(x_{t_i} \mid x_{t_{i-1}}).$$

We assume that

$$p_\theta(x_{t_i} \mid t_{i-1}) = \mathcal{N}(\mu_{\theta,i}, g_i I_d).$$

Then,

$$p_\theta(x_{[t',t]} \mid x_t = x) \propto \prod_{i=1}^{k} \exp\left(-\frac{\|\mu_{\theta,i} - (x_{t_i} - x_{t_{i-1}})\|^2}{2g_i^2}\right)$$

Therefore

$$\log p_\theta(x_{[t',t]} \mid x_t = x) = C + \sum_{i=1}^{k} \frac{\|\mu_{\theta,i} - (x_{t_i} - x_{t_{i-1}})\|^2}{2g_i^2}$$

where $C$ corresponds to the normalizing factor that is independent of $\theta$. Differentiating, we get that

$$\nabla_\theta \log p_\theta(x_{[t',t]} \mid x_t = x) = \sum_{i=1}^{k} \frac{\left(\mu_{\theta,i} - (x_{t_i} - x_{t_{i-1}})\right)^\top \nabla_\theta \mu_{\theta,i}}{g_i^2}$$

### B.2 Proof of Lemma 3.1

In what appears below, the expectation $\mathbb{E}[\cdot \mid x_t = x]$ is taken with respect to the distribution obtained by Eq. (7), namely, the backward SDE that corresponds to the function $s$, with the initial condition $x_t = x$. Similarly, $\mathbb{E}[\cdot \mid x_{t'}]$ is taken with the initial condition at $x_{t'}$. To prove the first direction in the equivalence, assume that Property 1 holds and our goal is to prove the two consequences as described in the lemma. To prove the first consequence, by the law of total expectation and by the fact that $x_t - x_{t'} - x_0$ is a Markov chain, namely, $x_0$ and $x_t$ are independent conditioned on $x_{t'}$, we obtain that

$$h(x,t) = \mathbb{E}[x_0 \mid x_t = x] = \mathbb{E}[\mathbb{E}[x_0 \mid x_{t'}] \mid x_t = x] = \mathbb{E}[h(x_{t'}, t') \mid x_t = x].$$

To prove the second consequence, by Property 1

$$h(x,0) = \mathbb{E}[x_0 \mid x_0 = x] = x_0.$$

This concludes the first direction in the equivalence.

To prove the second direction, assume that $h(x,t) = \mathbb{E}[h(x_{t'}, t') \mid x_t = x]$ and that $h(x,0) = x$ and notice that by substituting $t' = 0$ we derive the following:

$$h(x,t) = \mathbb{E}[h(x_0, 0) \mid x_t = x] = \mathbb{E}[x_0 \mid x_t = x],$$

as required.

## C  Additional Results

### C.1  Property Testing

**Discussion.**   We add some details about the plots:

- We note that the plots are not monotonic. This might be attributed to the model being closer to the ideal denoiser at certain noise levels and perhaps for having the capacity to correct previous mistakes.

- For each plot, we sample 32 images, and for each $t$ we get 32 stochastic samples conditioned on each one of the images and we measure the violation of MP. Standard deviation was small that it was not visible in the plot.

- $\sigma_t$ denotes the standard deviation of the noise at time $t$ and it is a monotonic function of $t$.

We underline that for the perfect denoiser, the error would have been $0$ everywhere. However, even after the MP regularization, the learning is not perfect hence there are errors.

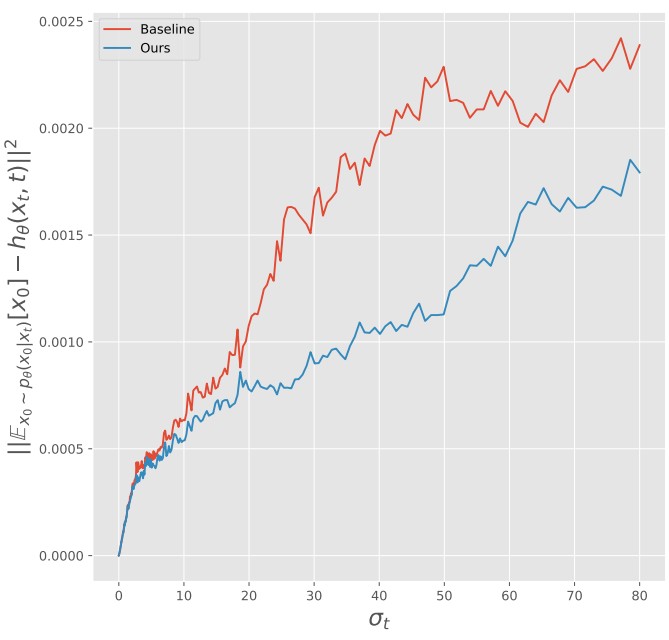

Figure 4: Consistency Property Testing on CIFAR-10. The plot illustrates how the Consistency Loss, $L_{t,t',x_t}^{\mathrm{CP}}$, behaves for $t' = 0$, as $t$ changes.

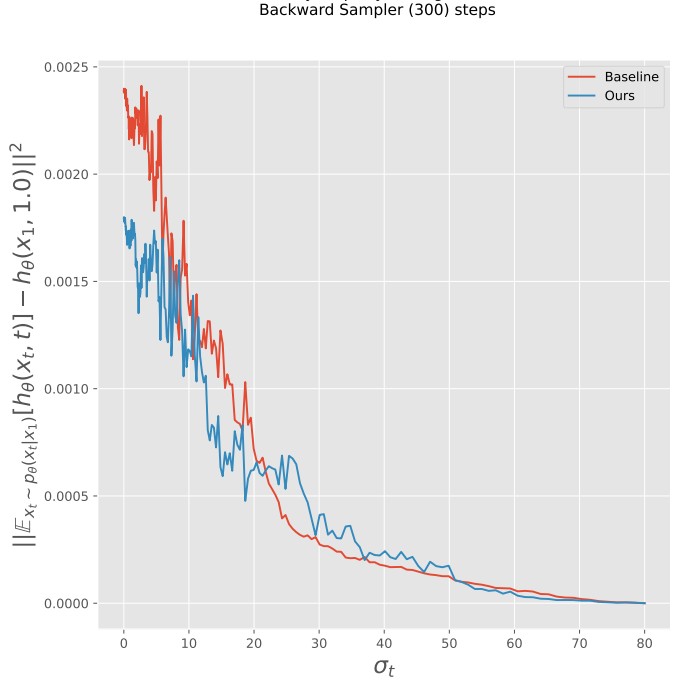

Figure 5: Consistency Property Testing on CIFAR-10. The plot illustrates how the Consistency Loss, $L_{t,t',x_t}^{\mathrm{CP}}$, behaves for $t = 0$, as $t'$ changes.

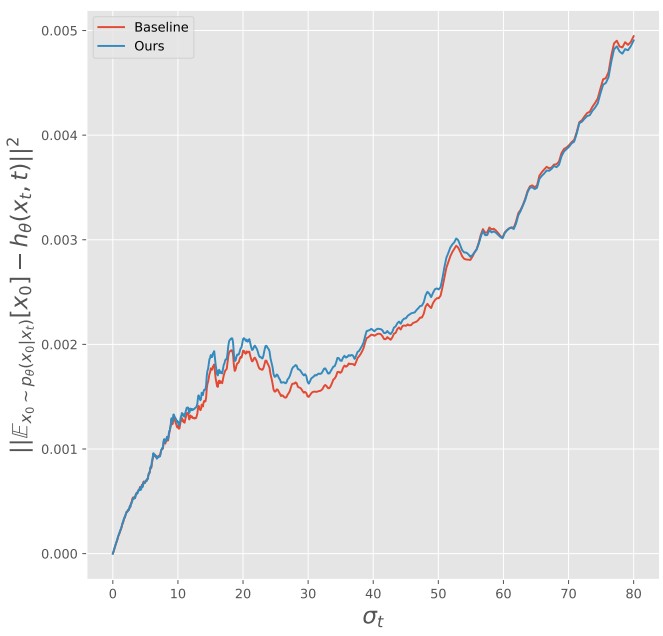

Figure 6: Consistency Property Testing on FFHQ. The plot illustrates how the Consistency Loss, $L^{\text{CP}}_{t,t',x_t}$, behaves for $t' = 0$, as $t$ changes.

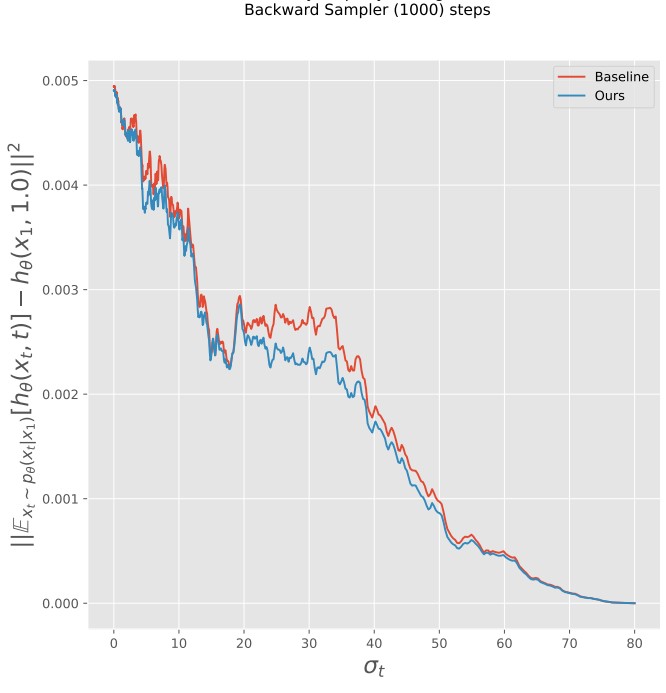

Figure 7: Consistency Property Testing on FFHQ. The plot illustrates how the Consistency Loss, $L^{\text{CP}}_{t,t',x_t}$, behaves for $t = 0$, as $t'$ changes.

## C.2 Uncurated Samples

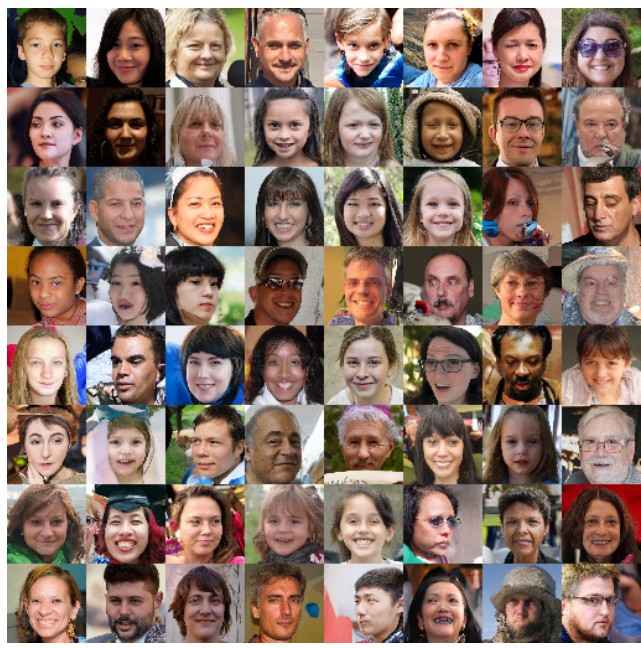

Figure 8: Uncurated generated images by our fine-tuned model on FFHQ. FID: 2.61, NFEs: 79.

# D Implementation details

---

**Algorithm 1** Consistent Diffusion Models (CDM) Training

---

**Input:** dataset $\mathcal{D}$, noise schedule $\{\sigma_t\}_0^T$, initial model parameters $\theta$, learning rate $\eta$, Consistency Property (CP) regularization strength $\lambda$, maximum distance $\epsilon$ between $t$ and $t'$, and step size $\Delta t$ for discretizing the Reverse SDE.

  **repeat**
      Sample $x_0 \sim \mathcal{D}$                                             $\triangleright$ (Clean image)
      Sample $\sigma_t$ uniformly from $\{\sigma_t\}_\epsilon^T$                     $\triangleright$ (Corruption level)
      Sample $x_t \sim \mathcal{N}(x_0, \sigma_t^2 I_d)$                       $\triangleright$ (Corrupted image)
      $L^{\text{SM}} \leftarrow ||h_\theta(x_t, t) - x_0||^2$              $\triangleright$ (Compute Score Matching loss)
      Sample $t' \sim \mathcal{U}[t - \epsilon, t]$               $\triangleright$ (Level with less corruption)
      **Stopgrad**
      Sample $x_{t'}^{(1)}, x_{t'}^{(2)}$ by running $\lfloor \frac{\epsilon}{\Delta t} \rfloor$ steps of the Reverse SDE $\triangleright$ (Sample less corrupted images)
      **Resumegrad**
      $L^{\text{CP}} \leftarrow \left( h_\theta(x_{t'}^{(1)}, t') - h_\theta(x_t, t) \right)^T \left( h_\theta(x_{t'}^{(2)}, t') - h_\theta(x_t, t) \right)$ $\triangleright$ Enforce CP with stochastic
gradient.
      $\theta \leftarrow \theta - \eta \nabla_\theta \left( L^{\text{SM}} + \lambda L^{\text{CP}} \right)$             $\triangleright$ (Update parameters)
  **until** convergence

---

# E Limitations

The capacity for generative models to exert consequential societal influence in myriad ways is undeniable, and it also brings along a multiplicity of inherent risks [37, 25, 26, 27]. These models, for instance, may be exploited to fabricate counterfeit images, and furthermore, they have the potential to intensify societal prejudices. This work does not seem to exert a direct influence on these particular biases. It is imperative to acknowledge that addressing such biases presents a substantial challenge.

