# OpenReview forum: "Consistent Diffusion Models: Mitigating Sampling Drift by Learning to be Consistent"
_NeurIPS.cc/2023/Conference — NeurIPS 2023 poster_

### Official Review · Reviewer_w9Z2 · 2023-07-06

**Soundness:** 4 excellent
**Presentation:** 3 good
**Contribution:** 2 fair
**Rating:** 5
**Confidence:** 4

**Summary:**

This paper introduced Martingale Property (MP) in diffusion model and the training strategy to apply this property to modify the score network outputs. In short, the martingale property states the condition that the SDE trajectory conditioned in a sample at time t is equal to the denoising function. After showing that MP transfers the score functions of small t to that of large t, this paper adds additional loss term that considers MP that the denoising function at time t' to be equal to that at time t, where t' < t. With this loss term added with the EDM model, the image generation quality in terms of FID is improved.

**Strengths:**

* This paper introduces a unified technique to improve the diffusion model with simple fine-tuning with short SDE trajectory starting from noisy data. Training with this new loss improves the sampling performance with some margin for CIFAR and AFHQ datasets.
* Compared to the concurrent work [1] that focused on the small-NFE performances, this directly learns "better" score function by reducing the learning variance, where the variance between x_t' and x_t is much smaller than that of x_0 and x_t.

**Weaknesses:**

* The FID performance gain is small, considering the additional training budget consumed.
* Theoretically, training solely with MDM loss term should show fair generation quality, since when the MP is satisfied, the model should be learned properly. However, the performance became worse, as Table 2 have shown.
* In the date of submission, there is a concurrent work [1] that also modifies the score network output by distilling knowledge from less noisy data.
* The second term in the equation after line 198 is not yet considered. Adding (even after some approximation) this term may further improve the method.

[1] Y. Song et al., "Consistency models", https://arxiv.org/abs/2303.01469

**Questions:**

* Is there any benchmarks when the MDM is trained longer without the DSM loss (extending the result in Table 2) than the result presented in the paper?
* Because of the second strength (variance reduction by reducing the noise from $x_0\to x_t$ to $x_{t'}\to x_t$) this work may perform better in larger datasets. Are there any additional results produced for higher-resolution datasets?

=====

Corrections.

Line 145: conditionning --> conditioning

Line 163: is satisfies --> satisfies

Line 203: sufficient --> is sufficient

Line 262: 64x64 --> $64\times64$

**Limitations:**

The authors adequately addressed the limitations of this paper.

---

> ### Author Rebuttal · Authors · 2023-08-09
>
> We thank a lot the Reviewer for their feedback! We are glad that the Reviewer appreciated certain aspects of our work. In what follows, we attempt to address some remaining concerns.
>
>
> > In the date of submission, there is a concurrent work [1] that also modifies the score network output by distilling knowledge from less noisy data.
>
> There are some similarities with the concurrent CM work and we will gladly cite it and discuss it in our next revision. The motivation is different: CM attempts to solve the reverse ODE in one step. We attempt to improve generation quality. The two methods are similar in that they enforce some property that the model should satisfy. In the case of the CM, the ODE solver should produce the same solution when evaluated at points that belong to the same trajectory. Our property is that on expectation the predictions should not be changing for points that have the same origin. One way to view it is that CM applies our MP condition but on the deterministic sampler (for which the expectation becomes the point itself).
>
> We think that the similarities between these two works should be celebrated: by applying the MP regularization to the deterministic and the stochastic sampler you get two complimentary, nice properties: faster sampling and improved quality.
>
> > The FID performance gain is small, considering the additional training budget consumed.
>
> While we agree that the FID benefit is relatively small, we also want to point out that improving the state-of-the-art in such standard datasets is extremely hard and the margins of improvement are already very thin.
>
> > Theoretically, training solely with MDM loss term should show fair generation quality, since when the MP is satisfied, the model should be learned properly. However, the performance became worse, as Table 2 have shown.
>
> This is not exactly true. Only training with MDM loss guarantees that we learn the score function of some distribution, but not necessarily the score function of the true distribution. Our theoretical results suggest that we can use the score-matching loss only for certain noise levels and the MDM loss everywhere else and then learn (in theory) the score of the data distribution. The mere fact that we can get a reasonably good performance by not using the score-matching term for the last $20$% of the diffusion trajectory (in the side of clean images) was actually surprising to us and we consider it a validation test for our theoretical result.
>
> > The second term in the equation after line 198 is not yet considered. Adding (even after some approximation) this term may further improve the method.
>
> That is true. However, backpropagating through the sampler is computationally intensive. The concurrent Consistency Models paper that the Reviewer brought to our attention also uses a stopgrad operation that has a similar effect.
>
> > Is there any benchmarks when the MDM is trained longer without the DSM loss (extending the result in Table 2) than the result presented in the paper?
>
> In all the results in the paper, the MDM models and the baseline models were trained for the same number of steps (200K steps, following the setup of the EDM paper). For all our experiments, the optimal performance was obtained before the 200K steps -- the performance flattens or even slightly deteriorates after a while. Hence, we do not have any reason to believe that the performance would improve further for either the baseline or our models.
>
> >  Are there any additional results produced for higher-resolution datasets?
>
> Unfortunately, the computing required to train diffusion models from scratch for higher resolutions becomes enormous. For reference, the official EDM repository states that training on ImageNet requires 32 A100 GPUs for 13 days. This is way beyond our computational budget. However, as the Reviewer suggests, it is quite possible that our method would give even more significant benefits for higher resolution datasets.
>
> Finally, we want to thank the Reviewer for carefully reading our paper and for finding our typos. We commit to fixing them in the camera-ready version of our work.
>
> If there are additional questions, we would be happy to answer them!

---

> > ### Comment · Reviewer_w9Z2 · 2023-08-19
> > **Response to the rebuttal**
> >
> > Thank you for the detailed response. I apologize to the authors that it took some moments to choose my circumstance for this paper after reading the rebuttal. While there is a concurrent work that also gives rise in some gain on the performance (even though the CM achieves fast sampling),
> > * This paper gives more concrete theoretical background why the reverse Martingale property may be used for marginalization of the score function over the dataset,
> > * Improved the FID performance gain, which is an extreme red ocean.
> > Even though there is a critical weakness that the concurrent (and previous work at the time of the submission), I am raising the score to borderline accept for the aforementioned reasons.

---

### Official Review · Reviewer_Qt1z · 2023-07-06

**Soundness:** 3 good
**Presentation:** 3 good
**Contribution:** 2 fair
**Rating:** 5
**Confidence:** 2

**Summary:**

Using approximate score functions and discretization can lead to compounding distribution shift as samples drift toward less likely regions of the training distribution. The authors propose to address this problem by enforcing that the learned denoiser satisfies an invariant they call the Martingale Property: $E_{gen} [x_0 | x_t = x] = h(x, t)$, where $h$ is the optimal (true) denoiser. The authors prove that if the learned denoiser satisfies MP and the score function is conservative, then there exists some underlying corresponding diffusion process; moreover, uniqueness holds. This implies that if in addition the learned denoiser matches the optimal denoiser at any time in a small region of the spatial domain, then the learned and optimal denoisers must match everywhere. Empirically, an extra loss term encouraging MP is added to the standard score matching loss, and results suggest this improves generated samples.

**Strengths:**

- The presentation of the theoretical results is clear.
- Experimental results are promising. Although MP is expensive to enforce exactly, the experiments demonstrate that the proposed MP loss is effective in practice.

**Weaknesses:**

- The main theorem (Thm. 3.2) does not seem to be very actionable. The results assume that MP holds exactly, which is an unreasonable assumption when considering diffusion model training in practice.
- Some of the theoretical results seem to have limited novelty. For example, uniqueness (Lemma A.9), which underpins the second claim of the main theorem, appears to be a standard argument.
- The experiments are promising but somewhat limited. Results are reported for only two datasets (including CIFAR10 cond vs. uncond) and the benefits of the MP loss is not very clear.

**Questions:**

- In practice, how well does MP violation correlate to poor image quality?
- How does enforcing the MP loss affect the image quality as NFE changes?

**Limitations:**

It could be helpful to clarify the relationship between MP and the consistency property from [1]. In practice, does enforcing the martingale property in expectation along generated trajectories behave similarly to the consistency loss, which is enforced pointwise along ODE trajectories?

---

> ### Author Rebuttal · Authors · 2023-08-09
>
> We thank the Reviewer for their feedback. We are happy that the Reviewer appreciated the theoretical and experimental aspects of our work.
>
> > The main theorem (Thm. 3.2) does not seem to be very actionable. The results assume that MP holds exactly, which is an unreasonable assumption when considering diffusion model training in practice.
>
> We agree that Theorem 3.2 is not actionable by itself. However, the theorem leads to our method (Section 4) which is very actionable since it gives a new loss function and then is used to train better diffusion models.
>
> While we agree that the MP assumption will not hold exactly, satisfying this property is easier compared to learning the score of the true distribution. Namely, satisfying the MP property implies that we have learned the score of some distribution, not necessarily the score of the data distribution. The whole training and sampling of diffusion models has been based on the assumption that the latter is perfectly learned. Even though this assumption does not hold exactly in practice, it has led to a useful framework for generative modeling. Theoretical works have modeled how errors caused by violation of this assumption propagate, e.g. see [1, 2]. We think that a similar analysis could be carried out for the case where MP is approximately satisfied. Yet, studying error propagation in PDEs requires a new analysis and we will leave it for future work.
>
> > Some of the theoretical results seem to have limited novelty. For example, uniqueness (Lemma A.9), which underpins the second claim of the main theorem, appears to be a standard argument.
>
> The goal of the theory section of this paper is to provide mathematical insight into the sampling drift problem. We develop it to suggest an algorithm to facilitate handling distribution shift in a principled fashion.
>
>
> > The experiments are promising but somewhat limited. Results are reported for only two datasets (including CIFAR10 cond vs. uncond) and the benefits of the MP loss is not very clear.
>
> We have experiments on CIFAR-10 (conditional and unconditional), AFHQ and FFHQ (finetuning). While we agree that the FID benefit is relatively small, we also want to point out that improving the state-of-the-art in such standard datasets is extremely hard and the margins of improvement are already very thin.
>
> > In practice, how well does MP violation correlate to poor image quality?
>
> That's a great question. In our loss function, we have two terms: the MP regularization and the score-matching loss. By increasing too much the weight of the MP term, we can even have detrimental effects in the generation quality -- MP (+ conservativeness) only ensures that we are learning some score function. However, for a properly chosen, fixed weight for the MP term, we get consistent performance benefits in generation quality when the MP loss drops. We will gladly discuss this in the paper and provide the training plots that demonstrate this.
>
>
> > How does enforcing the MP loss affect the image quality as NFE changes?
>
> Great question! The MP regularized models outperform the baselines in all NFE regimes. We observed that the performance gap is even larger for low NFEs (MP models seem more robust to decreasing the number of steps), but the optimal performance is obtained at the same number of NFEs for both the baseline and the regularized models.
>
> > It could be helpful to clarify the relationship between MP and the consistency property from [1]. In practice, does enforcing the martingale property in expectation along generated trajectories behave similarly to the consistency loss, which is enforced pointwise along ODE trajectories?
>
> There are some similarities with the concurrent CM work and we will gladly cite it and discuss it in our next revision. The motivation is different: CM attempts to solve the reverse ODE in one step. We attempt to improve generation quality. The two methods are similar in that they enforce some property that the model should satisfy. In the case of the CM, the ODE solver should produce the same solution when evaluated at points that belong to the same trajectory. Our property is that on expectation the predictions should not be changing for points that have the same origin. One way to view it is that CM applies our MP condition but on the deterministic sampler (for which the expectation becomes the point itself).
>
> We think that the similarities between these two works should be celebrated: by applying the MP regularization to the deterministic and the stochastic sampler you get two complimentary, nice properties: faster sampling and improved quality.
>
>
> [1] Sampling is as easy as learning the score
>
> [2] Restoration-Degradation Beyond Linear Diffusions: A Non-Asymptotic Analysis For DDIM-Type Samplers
>
>
> We hope that our response addresses the concerns of the Reviewer and that it can lead to an increase in the rating of our work. If there are any additional questions, we will remain available throughout the Paper Discussion period!

---

> > ### Comment · Reviewer_Qt1z · 2023-08-19
> >
> > Thanks to the authors for the detailed responses, and especially for the discussion of the similarities with the concurrent CM work. As mentioned in the rebuttal, it is interesting to see how CM and MP converge to a similar invariant idea from two different starting motivations,  and I look forward to seeing this discussion in the updated version of the manuscript.
> >
> > Based on the fact that the authors' theoretical analysis of sampling drift bears out empirically in (even slightly) improved FID scores in a competitive landscape, I am increasing my score to borderline accept. I think more experiments could be done to further support the authors' argument that MP leads to improved image quality (e.g. for a given dataset vary the MP regularization weight and show how FID scores change) but as the manuscript stands I lean toward accept.

---

### Official Review · Reviewer_G7yx · 2023-07-07

**Soundness:** 3 good
**Presentation:** 2 fair
**Contribution:** 3 good
**Rating:** 7
**Confidence:** 4

**Summary:**

Diffusion models are trained to denoise images that have been generated using a particular schedule of additive Gaussian noise; however, when sampling from a trained diffusion model, one applies a chain of denoising steps, each one operating on the output of the previous step. This may lead to sampling drift, where the distribution of the images at intermediate denoising steps differs from the distribution encountered during training. This sampling drift is hypothesized to be detrimental to the sampling process.

This paper proposes an approach to mitigate this drift, by enforcing a Martingale property (MP), which states that the denoising function $h_{\theta}(x, t)$ for all $t \in (0, 1]$ and $x \in \mathbb{R}^d$ outputs $h_{\theta}(x, t) = \mathbb{E}[x_0 \mid x_t = x]$, where this expectation is over the reverse diffusion process that starts from noisy example $x_t$ and runs the diffusion SDE, which applies the learned denoiser $h_{\theta}(x, t)$. They show that to enforce the MP, $h_{\theta}$ must satisfy a consistency property $h_{\theta}(x, t) = \mathbb{E}\_{h} \left[ h_{\theta}(x_{t'}, t') \mid x_t = x \right]$ for all $t > t'$, where the expectation is over all $x_{t'}$ sampled using the reverse diffusion process with denoiser $h_{\theta}$. In addition to the consistency property, they require the boundary condition $h(x, 0) = x$, such that at time $t=0$, the denoiser simply outputs the original input.

In order to encourage a denoiser to satisfy the MP, the authors propose a loss function that takes two successive points along the reverse diffusion trajectory, $x_t$ and $x_{t'}$, where $t'$ is slightly smaller than $t$, and minimizes the squared error between the denoiser output on those two points, $\mathcal{L}\_{t, t', x}(\theta) = \frac{1}{2} \left( \mathbb{E}\_{\theta}\left[ h_{\theta}(x_{t'}, t') \mid x_t = x \right] - h_{\theta}(x_t, t) \right)^2$. This term acts as a regularizer (with a weighting coefficient $\lambda$) on top of the standard denoising objective.

The paper presents results on several real-world datasets, including CIFAR-10, AFHQ, and FFHQ. Overall, it is not completely convincing that enforcing the Martingale property is worth the effort (of computing the datapoint pairs $(x_t, x_{t'})$ and potentially tuning the contribution of the new loss term). The results do not seem to improve dramatically when using the MP term.


**Strengths:**

* The paper addresses an important problem with good motivation, to mitigate drift when sampling from a trained diffusion model. The idea to enforce the Martingale property is very nice, and the authors develop this idea into a practical method that adds an additional loss term to the standard denoising objective.

* The proposed approach is theoretically justified, and fairly intuitive. The authors give a thorough discussion of the necessary assumptions and theoretical properties of their approach.

* The authors apply the Martingale property regularizer to train diffusion models from scratch as well as to fine-tune pre-trained diffusion models. They evaluate the approach on three datasets---CIFAR-10, AFHQ, and FFHQ---and they show slight improvements over two baselines in terms of FID scores.

* The authors also verify that their approach decreases the Martingale loss, which measures consistency in the denoised outputs, comparing this loss between a baseline model and one trained with the Martingale regularizer.

**Weaknesses:**

* The proposed framework has some similarities to Consistency Models (Song et al., 2023) and it would be helpful if the authors could add more discussion of this in the paper. I don't think the two approaches are identical. The core similarities are in the reverse-Martingale property, which looks basically the same as the consistency property for CM, that states that the output of the function should be equivalent whether it is computed on $x_t$ or $x_{t'}$ for any $t' < t$. Also, both CM and this work have a similar boundary condition for $t=0$. The motivations for the approaches are different, as CM aims to produce a 1-step sampler that maps directly from noise to a clean image, while the Martingale property is used to mitigate sampling drift to improve sample quality when using standard sequential sampling chains. Both approaches sample two points along the denoising trajectory, $x_t$ and $x_{t'}$, which are used to compute a consistency-like loss.

* One apparent difference between MP and CM is that the MP uses an expectation over samples from the reverse diffusion process for a single input $x$. Also, MP seems to apply the same parameters $\theta$ to both parts of the consistency function, while CM uses an exponential moving average of the parameters for one part? In MP, what is the number of samples used to approximate the expectation in the first part, $\mathbb{E}\_{\theta}[h_{\theta}(x_{t'}, t') \mid x_t = x]$?

* The paper should specify clearly what the difference is between $\bar{B}_t$ and $B_t$ in equations 3 and 4.

* In L33, why has the notation switched to using superscripts $p^*_0$? Does this denote anything different from the non-superscripted version $p_0$ used just before this sentence?

* In L38, I think it would be clearer if the learned score function were denoted by writing its parameters $s_{\theta}(x, t)$ rather than just $s(x, t)$, to clarify how it differs from the true score function $s^*(x, t)$. Otherwise, this distinction is subtle and unclear.

* Equation numbers need to be added to the two equations in the introduction (page 2). Also, it is confusing that the order of the left- and right-hand sides are flipped between the first and second equation on page 2. I think it would be clearer for the second equation to be $h(x, t) = \mathbb{E}[x_0 \mid x_t = x], \forall t \in [0, 1] \forall x$.

* The superscripts used to denote the loss functions $L^1_{t, x_t, x_0}$ and $L_{t, t', x}^2$ are confusing because they look like they would represent $L_1$ and $L_2$ losses, respectively, while they are actually just indexes for the first and second loss terms. It would be better to rename them $L^{\text{SM}}$ for the score matching loss and $L^{\text{MP}}$ for the Martingale property loss.

* The title, caption, and labels are too small in Figures 1(a,b). Why are the "baseline" and "Ours" curves following the same pattern (e.g., the same fluctuations)? Why are the curves non-monotonic? Are the plots shown for one example or are they an average over many examples? Could the authors report min/max values or the standard deviation over different examples for these curves? What do the values of $\sigma_t$ from 0 to 80 represent?

* Why is the FFHQ dataset not included in Table 1?

**Minor**
* L9 typo: "describes conservative" --> "describes a conservative"

* L14 typo: "in CIFAR-10" --> "on CIFAR-10"

* In Eq. 1, explain the notation $\oplus$ as it is fairly non-standard.

* The conventional notation for normal distributions is $\mathcal{N}$, not $N$.

* L24: "$\sigma_t$ is an increasing function" --> $\sigma_t$ is a value, not a function, so instead one can say that $\sigma_t$ is given by an increasing function?

* L44: It is strange to say "Question 1" when there are no other questions.

* L84: "The formal statement is summarized as follows below: Theorem 1.1 (informal)" --> The "formal" statement immediately says "informal."

* In Eq. 2, why use this strange notation, rather than following the standard convention of using subscripts of $\mathbb{E}$, that is $\mathbb{E}_{x_0 \sim p_0, x_t \sim \mathcal{N}(x_0, \sigma_t^2 I_d)}$?

* L41 typo: "the larger is also the error" --> "the larger the error"

* In L52, "relates multiple inputs to $s(\cdot, \cdot)$" --> Before seeing the final consistency objective, it is unclear what is meant by this.

* L58 typo: "this phenomena" --> "this phenomenon"

* L59 typo: "called optimal denoiser" --> "called the optimal denoiser"

* L64 typo: "technique via score-matching" --> "technique of score-matching"

* L167 typo: "can be relieved" --> "can be relaxed"

* L182: I do not think that the term VE-SDE or VP-SDE have been defined yet, before they are used.

* L188 typo: "trajectories:" --> "trajectories."

* L195 typo: "$x_{t'}$ and $x_{t'}$" --> these are the same.

* In the last equation on page 5, it is hard for the reader to see which is the first and second term of the summand. This equation should be numbered, and the summand terms could be labeled explicitly, so that it's clear which one is used in practice.

* Typo in equation at the bottom of page 5: there is an extra bracket in $\mathbb{E}_{\theta}[]$.

* L207 typo: "choises" --> "choices"

* L228 typo: "droped" --> "dropped"

* L262 typo: "$64x64$" --> "$64 \times 64$"

* Table 1 has a significant amount of empty space, as the last 4 rows for each dataset block are empty in 6 out of 7 columns. Also, most of the numbers in the final column come from other sources.


**Questions:**

* An important point regarding notation is that throughout the paper, it is unclear what certain expectations are taken with respect to. In the two equations on page 2, neither expectation has any subscript to indicate this, yet it is very important (it is written in the text below, but it is also crucial to be precise in the equations). Also, it is slightly confusing that the expectations are not written consistently: in some places, there is no subscript, in others the subscript is $\mathbb{E}_h$, and in others it is $\mathbb{E}\_{\theta}$. I am not sure whether any of these makes it clear that the sample $x_0$ is obtained from the learned reverse process defined in terms of $h$? Because $\mathbb{E}_h$ implies that the expectation is over all possible functions $h$. Would it make sense to use alternative notation like $\mathbb{E}\_{x \sim \text{Reverse}(x_t, h\_{\theta})}$?

* It is not clear how representative the examples in Figure 3 are, because it may be possible to find samples that are improved by MP and others that are not; did MP regularization help with the majority of images?

* It would be good to add a discussion on why the second term of the last equation on page 5 can be dropped?



**Limitations:**

* As the authors acknowledge, they do not verify the assumption that the learned vector field is conservative, which is necessary for the theoretical results.

* The authors do not provide an algorithm box to state the approach formally (not in the main paper nor in the appendix).

---

> ### Author Rebuttal · Authors · 2023-08-09
>
> We thank the reviewer for the extremely detailed and constructive feedback. We will incorporate all the thoughtful comments regarding the presentation of our work.
>
> We agree that there are some similarities with the concurrent CM work and we will gladly discuss it. The motivation is different: CM attempts to solve the reverse ODE in one step. We attempt to improve generation quality. The two methods are similar in that they enforce some property that the model should satisfy. In the case of the CM, the ODE solver should produce the same solution when evaluated at points that belong to the same trajectory. Our property is that on expectation the predictions should not be changing for points that have the same origin. One way to view it is that CM applies our MP condition but on the deterministic sampler (for which the expectation becomes the point itself).
>
> We think that the similarities between these two works should be celebrated: by applying the MP regularization to the deterministic and the stochastic sampler you get two complimentary, nice properties: faster sampling and improved quality.
>
> > Also, MP seems to apply the same parameters $\theta$ to both parts of the consistency function, [...]
>
> This is true, we do not use EMA.
>
> > In MP, what is the number of samples used to approximate the expectation in the first part, $\mathbb{E}_{\theta}[h_{\theta}(x_{t'}, t') \mid x_t = x]$?
>
> We can get an unbiased estimator of the gradient, by only using two samples to estimate the expectation. This is what we do in all the experiments to avoid computational overhead (lines 195-197).
>
> > The paper should specify clearly what the difference is between $B_t$ and $\bar{B}_t$ [...]
>
> $B_t$ is a Brownian motion whereas $\bar{B}_t$ is a Brownian motion that runs backwards in time. We will clarify it in the paper.
>
> > In L33, why has the notation switched to using superscripts $p_0^*$?
>
> We use the notation $p_0^*$ when we discuss a distribution that we would like to learn whereas we use $p_0$ when the discussion is only about the mathematical properties of diffusion processes.
>
> > In L38, I think it would be clearer if the learned score function were denoted by writing its parameters [...]
>
> We will update the text accordingly to make this distinction clear.
>
> > Why are the "baseline" and "Ours" curves following the same pattern (e.g., the same fluctuations)? [...]
>
> This figure shows how much the Martingale Property is violated as we keep $t’=0$ and we change $t$ from $0$ to $T$. $\sigma_t$ denotes the standard deviation of the noise at time $t$ and it is an increasing function of $t$. For the perfect denoiser, the error would have been $0$ everywhere. However, even after the MP regularization, the learning is not perfect and as the distance of $t$ and $t’$ increases the property gets violated. We should not necessarily expect this function to be monotonic. The non-monotonic increase might be attributed to the model being closer to the ideal denoiser at certain noise levels and perhaps having the capacity to correct previous mistakes. The message of the plot is that the MDM models violate MP less. The fluctuations are similar. This could be related to the scheduling of the noise; both models might struggle to maintain MP at the same levels. Another explanation could be that we are adding the same noise realization. For this plot, we first sample $32$ images and for each $t$ we get $32$ stochastic samples conditioned on each one of the images and we measure the violation of MP. Standard deviation was so small that it was not visible in the plot. We will include these details in our next revision.
>
> > Why is the FFHQ dataset not included in Table 1?
>
> Following the Reviewer’s recommendation, we will include it in Table 1 with the rest of the results and we will name it as FFHQ (finetuning) to avoid confusion.
>
> > As the authors acknowledge, they do not verify the assumption that the learned vector field is conservative, which is necessary for the theoretical results.
>
> We recently came across [this](https://arxiv.org/abs/2209.12753) paper that investigates how to make diffusion models satisfy the conservative property. We will make sure to add this reference.
>
> > The authors do not provide an algorithm box to state the approach formally (not in the main paper nor in the appendix).
>
> We thank the Reviewer for the suggestion. We include an algorithm box in the one page PDF that accompanies this rebuttal, we will also include it in the main paper in the camera-ready version of our submission.
>
> > An important point regarding notation is that throughout the paper, it is unclear what certain expectations are taken with respect to [...].
>
> In the first Eq. on page 2, the conditional expectation is with respect to the joint distribution between $x_0$ that is sampled from $p_0^*$ and $x_t$ that is obtained by adding noise: $x_t = x_0 + \sigma_t \eta$. In the second Eq., one again computes the conditional expectation of $x_0$ given $x_t = x$, but here, the conditional expectation is with respect to the joint distribution where $x_t$ is fixed and $x_0$ is sampled by running the differential equation described by Eq. (3) backward in time.
> We will improve the readability of the paper according to the suggestions.
>
> > It is not clear how representative the examples in Figure 3 are [...]
>
> We generated 64 images from the baseline and our method and by inspection we found that we considerably improved 14 of them. For the rest of the samples we did not see considerable differences.
>
> > It would be good to add a discussion on why the second term of the last equation on page 5 can be dropped?
>
> We drop it because backpropagating through the sampler is computationally intensive. We note that CM does a similar stopgrad operation.
> Intuitively, dropping it might not matter much because of the score-matching term already giving sufficient signal to improve the Martingale loss. We will add this discussion to the paper.

---

> > ### Comment · Reviewer_G7yx · 2023-08-17
> > **Response to Rebuttal**
> >
> > I read the other reviews and the authors' rebuttals. The authors have answered all my questions and addressed my concerns. I think they offer a valid contribution to the field, and consequently I raise my score to 7.

---

### Official Review · Reviewer_UfbR · 2023-07-07

**Soundness:** 3 good
**Presentation:** 3 good
**Contribution:** 3 good
**Rating:** 7
**Confidence:** 3

**Summary:**

This paper proposed a new method to address the issue of imperfect score-matching in diffusion models by enforcing a Martingale Property (MP) during training. This MP property ensures that the model's predictions on its self-generated data are consistent with the generated outputs. The key contributions include the identification of an invariant property - the denoiser satisfies MP property - that any perfectly trained model should satisfy, proof of this principle is provided. Further, the paper introduces a novel training objective that enforces MP, optimizing the network to consistently predict data points from the learned distribution. Empirically, the paper showcases that using the proposed objective, paired with the original DSM loss, improves the generation quality in both conditional and unconditional generation contexts, validated on CIFAR-10, AFHQ, and FFHQ datasets.

**Strengths:**

The paper is well organised and presented, easy to follow.

The work is novel, enforcing the MP property cast the drift issue into a relatively straightforward question. The methodology is appropriately described and presented and should be of interests to researchers in diffusion model field.

In my view, general clarity and quality of this paper is good, mathematical formulas are clear and the claims are generally supported by the results.

**Weaknesses:**

It seems too much space was occupied by background introduction in the first 3.5 pages, might be good to further compress them.

The objective function presented in Section 4 makes sense if one follows the paper from the beginning, but might be a bit difficult for quick interpretation and implementation, would be nice if a further simplified version (or even a special case) under some certain conditions can be presented

Can Figure 1 be combined as a single figure? - it doesn't add too much information to show them in two separate figures.

**Questions:**

1. Could the author provides a bit more detailed explanations to link the results in Table 1 (CIFAR10 parts) and Table 2?

2. Is it possible to provide results using metrics other than FID, such as MMD or JS etc to evaluate the data distribution from a different perspective?

**Limitations:**

Limitations were clearly stated by the authors and the discussion of future work is sensible.

---

> ### Author Rebuttal · Authors · 2023-08-09
>
> We thank a lot the Reviewer for appreciating the novelty, the theoretical and the empirical results and the presentation of our work!
>
> We will compress the background section and combine Figures 1a, 1b into a single Figure as recommended by the Reviewer.
>
> > The objective function presented in Section 4 makes sense if one follows the paper from the beginning, but might be a bit difficult for quick interpretation and implementation, would be nice if a further simplified version (or even a special case) under some certain conditions can be presented
>
> That’s a fair comment, we thank the Reviewer for raising it. In the one page PDF response that accompanies this rebuttal, we include the Algorithm we use in practice to train models with our MP regularization.
>
> We will make this Section more user-friendly by starting right away with this algorithm and by describing step by step how the method works. We will briefly give some intuition of what this loss is doing, which is to make sure that the network's prediction remains on average the same as we progress along the trajectories of the reverse SDE. We will abstract away the calculations needed to derive this (and point to the Appendix for the detailed calculations).
>
> > Could the author provides a bit more detailed explanations to link the results in Table 1 (CIFAR10 parts) and Table 2?
>
> Absolutely! The results in Table 1 are obtained by training with the loss function in Section 5, i.e. with $L_{\lambda}^{\mathrm{ours}}(\theta)$. This loss has two terms, the standard denoising score matching term and the MP regularization. This is the recommended way to train diffusion models with MP regularization and it leads to improved performance in image generation compared to the baseline that only uses the denoising score matching loss.
>
> Table 2 summarizes the results of an ablation study in support of our Theorem 3.2. According to this theorem, if we can learn the score perfectly only for some noise level (indexed by $t$), we satisfy the MP property everywhere and we have learned a conservative vector field, then we know the score perfectly everywhere.
>
> In order to test whether this theoretical result captures (at least partially) practical behaviour of diffusion training, we train using the DSM objective only for $80$% of the diffusion time and we do not use it at all for last $20$% of the diffusion trajectory (on the side of clean images). We show that without the MP regularization for the last $20$% of the diffusion trajectory (MDM with early stopped sampling) the FID increases a lot (i.e. performance drops significantly). However, using the MP regularization for the last $20$%, the performance doesn’t drop significantly compared to using both the MP regularization and the DSM loss for the whole trajectory (as we did in Table 1). We hope this clarifies things and we will update the text accordingly to make this clearer.
>
> > Is it possible to provide results using metrics other than FID, such as MMD or JS etc to evaluate the data distribution from a different perspective?
>
> We followed the evaluation protocol (and the code) of the EDM paper that does not report such metrics. We will include those numbers for both the baseline and our method in the camera-ready version of our work. We thank the Reviewer for the suggestion!
>
> We will be happy to answer additional questions if any throughout the Paper Discussion period. We thank again the Reviewer for their constructive feedback!

---

> > ### Comment · Reviewer_UfbR · 2023-08-18
> >
> > Thanks for the author's response, it does answer most of my curiosities. I am happy to keep the current rating.

---

### Author Rebuttal · Authors · 2023-08-09

We thank the Reviewers for their constructive feedback. We reply separately to each of the Reviewers. Following the recommendation of Reviewer G7yx, we also include a formal statement of our training algorithm, given in the one-page PDF accompanying this rebuttal.

We thank again the Reviewers and we will gladly answer additional questions, if any!

---

### Decision · Program_Chairs · 2023-09-21

**Decision:**

Accept (poster)

**Comment:**

The authors develop a technique to address errors in generation in a diffusion model translating to evaluations of the score model outside of where it was trained. The technique develops a new loss term to enforce a martingale property of the reverse diffusion. When combined with existing loss functions, the approach leads to improved results. After all of the discussion, all reviewers are positive.

I'd also suggest the authors change the acronym from MDM to something else as MDM was used recently in a diffusion paper for the term multivariate diffusion models at ICLR 2023 (https://openreview.net/pdf?id=osei3IzUia).